# DISCOVERING MESSAGE PASSING HIERARCHIES FOR MESH-BASED PHYSICS SIMULATION

## ABSTRACT

Graph neural networks have emerged as a powerful tool for large-scale mesh-based physics simulation. Existing approaches primarily employ hierarchical, multi-scale message passing to capture long-range dependencies within the graph. However, these graph hierarchies are typically fixed and manually designed, which do not adapt to the evolving dynamics present in complex physical systems. In this paper, we introduce a novel neural network named DHMP, which learns **D**ynamic **H**ierarchies for **M**essage **P**assing networks through a differentiable node selection method. The key component is the *anisotropic* message passing mechanism, which operates at both intra-level and inter-level interactions. Unlike existing methods, it first supports directionally non-uniform aggregation of dynamic features between adjacent nodes within each graph hierarchy. Second, it determines node selection probabilities for the next hierarchy according to different physical contexts, thereby creating more flexible message shortcuts for learning remote node relations. Our experiments demonstrate the effectiveness of DHMP, achieving 22.7% improvement on average compared to recent fixed-hierarchy message passing networks across five classic physics simulation datasets.

## 1 INTRODUCTION

Simulating physical systems with deep neural networks has achieved remarkable success due to their efficiency compared with traditional numerical solvers. Graph Neural Networks (GNNs) have been validated as a powerful tool for mesh-based physical scenarios, such as fluids and rigid collisions (Wu et al., 2020). The primary mechanism driving the GNN-based models is message passing (Sanchez-Gonzalez et al., 2020; Pfaff et al., 2021; Allen et al., 2023). In this process, time-varying physical quantities are encoded within the mesh structure at each time step and are updated by aggregating information broadcast from neighboring nodes. These existing methods generally rely on local message passing, limiting their ability to propagate influence over long distances. A common solution involves using multi-scale graph structures to facilitate direct information shortcuts between distant nodes (Lino et al., 2022; Cao et al., 2023; Yu et al., 2024; Han et al., 2022; Fortunato et al., 2022).

However, as illustrated in Table 1, these approaches depend on heuristic methods to create coarser message passing structures with predefined graphs (Cao et al., 2023; Yu et al., 2024) or downsample the nodes based on spatial proximity (Lino et al., 2022) where hierarchies are preprocessed in one pass before training. These fixed graph hierarchies over the entire physical sequence do not account for the diverse range of physical contexts; while in practical systems like turbulence, despite identical boundary conditions, even minor changes in initial conditions can lead to significant differences in subsequent dynamics. Moreover, the spatial correlations in a physical process can evolve over time, making static GNN hierarchies insufficient for accommodating the time-varying node interactions.

To tackle these challenges, we propose a novel approach named Dynamic Hierarchical Message Passing (DHMP), which constructs context-aware and temporally evolving graph hierarchies based on the original mesh topology and the input physical quantities. The key insight is to develop a differentiable node selection method that allows for flexible modeling of node interactions. This is technically supported by the proposed *anisotropic message passing*, which aggregates the neighboring features to the central node in a directionally non-uniform manner, predicting its downsampling probabilities as a node within the coarser graph level. We then employ Gumbel-Softmax sampling to create a differentiable approximation of the hard sampling for the downsampled graph.

Table 1: Comparison of mesh-based simulation models. *Dynamic hierarchy* refers to hierarchical graph structures evolving over time. *Context-aware* indicates that the graph structures are determined by the physical inputs. *Prop.* denotes different feature propagation mechanisms.

| Model | Hierachical | Dynamic Hierarchy | Context-Aware Hierarchy | Anisotropic Intra-level Prop. | Learnable Inter-level Prop. |
|---|---|---|---|---|---|
| MGN (2020) | ✗ | - | - | ✗ | - |
| BSMS-GNN (2023) | ✓ | ✗ | ✗ | ✗ | ✗ |
| Lino *et al.*(2022) | ✓ | ✗ | ✗ | ✗ | ✓ |
| DHMP | ✓ | ✓ | ✓ | ✓ | ✓ |

The anisotropic message passing mechanism not only adaptively creates multi-scale graph structures but also enables learned directionally non-uniform importance weights to facilitate both intra-level and inter-level propagation of dynamic information. As shown in Table 1, existing approaches perform isotropic feature aggregation within intra-level transition, assuming equal contributions from neighboring nodes, which may overlook the directional nature of physical processes. While some methods employ attention mechanisms to replace isotropic intra-level propagation (Dwivedi & Bresson, 2020; Janny et al., 2023; Yu et al., 2024; Han et al., 2022), our approach demonstrates advantages in computational efficiency. Furthermore, existing models generally rely on unlearnable importance weights to transfer information across hierarchical levels. In contrast, the inter-level aggregation weights in DHMP are data-specific and time-varying, effectively harnessing the anisotropic nature of our message passing mechanism to enhance multi-scale modeling flexibility.

Overall, our contributions are summarized as follows:

- We present DHMP, a new method that constructs dynamic hierarchies via a differentiable node selection process, enabling context-aware modeling of hierarchical structures for physics simulations.
- As a key component in DHMP, the proposed anisotropic message passing enables learnable, non-uniform intra-level and inter-level feature propagation, significantly enhancing model performance.
- DHMP achieves a **22.7% promotion** on average across four standard benchmarks, compared with fixed-hierarchy models. It is also shown to generalize well to test cases with time-varying mesh structures (Table 3), unseen resolutions (Table 4), and out-of-distribution dynamics (Table 5).

## 2 PRELIMINARIES

**Message passing.** We consider simulating mesh-based physical systems, where the task is to predict the dynamic quantities of the mesh at future timesteps given the current mesh configuration. A mesh-based system is represented as a bi-directed graph $\mathcal{G} = (\mathcal{V}, \mathcal{E})$[1], where $\mathcal{V}$ and $\mathcal{E}$ denote the set of nodes and edges, respectively. *Message passing neural networks* (MPNNs) compute the node representations by stacking multiple message passing layers of the form:

$$\text{Edge update: } \hat{\mathbf{e}}_{ij} = \phi^e(\mathbf{e}_{ij}, \mathbf{v}_i, \mathbf{v}_j); \text{ Node update: } \hat{\mathbf{v}}_i = \phi^v(\mathbf{v}_i, \psi(\{\hat{\mathbf{e}}_{ij} \mid \forall j, e_{ij} \in \mathcal{E}\})), \quad (1)$$

where $\mathbf{v}_i$ is the feature of node $v_i \in \mathcal{V}$ and $\psi$ denotes a ***non-parametric*** aggregation function. The function $\phi^e$ updates the features of edges based on the endpoints, while $\phi^v$ updates the node states with aggregated messages from its neighbors. In existing mesh-based simulation methods, multi-layer perceptrons (MLPs) with residual connections are commonly employed for $\phi^e(\cdot)$ and $\phi^v(\cdot)$, with the aggregation function $\psi(\cdot)$ being defined as the sum of edge features. Notably, since the aggregation function treats all neighbors equally, the contributions from neighboring nodes may be averaged out, and the repeated message-passing process can further dilute distinctive node features. This issue is exacerbated in dynamic physical systems, where transferring directed patterns is crucial.

**Hierarchical MPNNs.** To facilitate long-range modeling, hierarchical MPNNs process information at $L$ scales by creating a graph for each level and propagating information between them (Lino et al., 2022; Fortunato et al., 2022; Cao et al., 2023; Yu et al., 2024). Let $\mathcal{G}_1 = (\mathcal{V}_1, \mathcal{E}_1)$ represent the graph structure at the finest level, *i.e.*, the input mesh. The lower-resolution graphs $\mathcal{G}_2, \mathcal{G}_3, \ldots, \mathcal{G}_L$, with $|\mathcal{V}_1| > |\mathcal{V}_2| > \ldots > |\mathcal{V}_L|$, contain fewer nodes and edges, which allows for more efficient

---

[1]*Bi-directed* means each original undirected edge is represented twice in $\mathcal{G}$: if there is an edge between $i$ and $j$, it is represented as two directed edges $i \rightarrow j$ and $j \rightarrow i$. Each node has a self-loop.

feature propagation over longer physical distances with certain propagation steps. The typical process for constructing multi-scale structures primarily involves downsampling and upsampling between adjacent graph hierarchies. Downsampling reduces the number of nodes while upsampling transfers information from a lower-resolution graph to a higher-resolution one. The downsampling operation can be broken down into two steps:

- SELECT: Nodes are selected from the current graph structure $\mathcal{G}_l$ to create a new, coarser graph $\mathcal{G}_{l+1}$. Various criteria for node selection (Diehl, 2019; Ying et al., 2018; Lino et al., 2022) can be applied to form $\mathcal{V}_{l+1}$. The edges $\mathcal{E}_{l+1}$ in $\mathcal{G}_{l+1}$ are constructed by connecting the selected nodes based on the original edges $\mathcal{E}_l$. However, this process can sometimes lead to loss of connectivity and introduce partitions (Gao & Ji, 2019; Lee et al., 2019; Cao et al., 2023). To mitigate this, connectivity in $\mathcal{E}_{l+1}$ can be strengthened by adding $K$-hop edges.

- REDUCE: The features of the nodes in $\mathcal{V}_{l+1}$ are aggregated from their corresponding neighborhood features in the finer graph $\mathcal{G}_l$.

The upsampling process is represented by the EXPAND, the inverse of the REDUCE function, which aggregates information from the coarser level back to the finer level. Most previous work generates coarser graphs for each sequence either by using numerical software or by downsampling the input mesh through heuristic pooling strategies. This process occurs during the data preprocessing stage, enabling the preprocessed hierarchy of the same input mesh topology to be reused across various initial conditions and different time steps.

## 3 METHOD

In this section, we introduce the Dynamic Hierarchical Message Passing Networks (DHMP), a fully differentiable model that learns to dynamically generate coarser graphs over the sequence while simultaneously learning to simulate the physical system over the learned hierarchical graphs.

### 3.1 OVERVIEW

Figure 1 demonstrates an overview of the proposed model, which operates in an *encode-process-decode* pipeline. The encoder first maps the input field to a latent feature space $\mathbf{V}_1 = \{\mathbf{v}_i | v_i \in \mathcal{V}_1\}$ at the original mesh resolution. Subsequently, we model the physical dynamics across the learned multi-scale graph hierarchies with adaptive graph structures. To enhance the propagation of long-term dependencies between distant nodes, we propose an *anisotropic message passing* (AMP) mechanism, which is largely inspired by the directed nature of significant dynamic patterns.

In Section 3.2, we present the details of the AMP layer. In Section 3.3, we discuss the approach for learning context-aware graph hierarchies. In Section 3.4, we describe the inter-level downsampling and upsampling processes that incorporate AMP-based feature propagation. Finally, in Section 3.5, we outline the implementation details and hyperparameter choices.

### 3.2 ANISOTROPIC MESSAGE PASSING

We introduce the AMP layer, which facilitates information propagation both within and between graph hierarchies, enabling DHMP to effectively capture local and long-range dependencies simultaneously.

As shown in Eq. (1), a common method in mesh-based simulation is to use the summation aggregation function for node update: $\hat{\mathbf{v}}_i = \phi^v\left(\mathbf{v}_i, \sum_{v_j \in \mathcal{N}_{v_i}} \hat{\mathbf{e}}_{ij}\right)$, where $v_j \in \mathcal{N}_{v_i}$ denotes a neighboring node of $v_i$ in the graph. Using the summation aggregation has two drawbacks: i) it can excessively smooth the neighboring features, potentially failing to capture intricate local relations, as discussed in previous literature (Alon & Yahav, 2021; Dong et al., 2023; Dwivedi et al., 2022), and ii) it does not account for the directed nature that can be inherent in physics scenarios.

To address these issues, we propose the AMP layer, which employs a more flexible aggregation function to facilitate anisotropic feature propagation within each message-passing hierarchy. Instead of directly summing the edge features, AMP exploits learnable parameters $\phi^w : \mathbb{R}^{F^e} \to \mathbb{R}$ to predict the importance weight of edge feature $\hat{\mathbf{e}}_{ij}$ to node $v_i$:

$$w_{ij} = \phi^w(\mathbf{e}_{ij}, \mathbf{v}_i, \mathbf{v}_j). \tag{2}$$

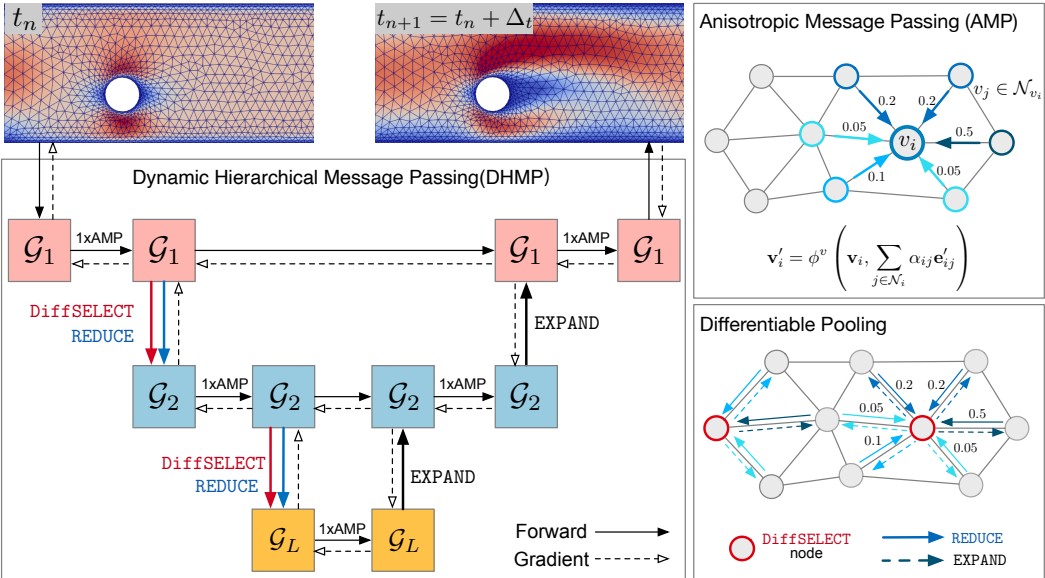

Figure 1: In DHMP, physical dynamics is modeled on multiple graph resolutions with adaptive structures, $\mathcal{G}_1, \mathcal{G}_2, \ldots, \mathcal{G}_L$, and are processed using their respective AMP layers. The `DiffSELECT` operation performs differentiable pooling to create coarser graphs with learnable downsampling probabilities. `REDUCE` and `EXPAND` integrate inter-level information using learned feature aggregation weights over the neighboring nodes. DHMP is trained end-to-end with one-step supervision.

To ensure that the coefficients are easily comparable across different nodes, we normalize them using the softmax function across all choices of $j$:

$$\alpha_{ij} = \text{softmax}_j \left( w_{ij} \right) = \frac{\exp \left( w_{ij} \right)}{\sum_{k \in \mathcal{N}_i} \exp \left( w_{ik} \right)}. \tag{3}$$

The normalized coefficients are used to compute a linear combination of the corresponding edge features. This linear combination serves as the final input for the node update function $\phi^v$ given node feature $\mathbf{v}_i$:

$$\hat{\mathbf{v}}_i = \phi^v \Big( \mathbf{v}_i, \sum_{v_j \in \mathcal{N}_{v_i}} \alpha_{ij} \hat{\mathbf{e}}_{ij} \Big). \tag{4}$$

Unlike traditional MPNNs with non-parametrized aggregation functions, the proposed AMP layer allows for the implicit assignment of varying contribution weights to the updated edge features within the same neighborhood. Analyzing the learned direction-specific weights in AMP provides additional benefits for interoperability. AMP also differs from the Graph Convolutional Networks (GCNs) (Niepert et al., 2016) and attention-based GNNs (Veličković et al., 2018): while these methods model aggregation by assigning weights to node features, AMP emphasizes weighting edge features which contain relative distance offsets. These edge features provide direct information about node positions, making them essential for capturing spatial relationships and enhancing generalization.

### 3.3 Differentiable Multi-Scale Graph Construction

With the AMP layer functioning within each graph level, local dependencies are effectively propagated throughout the high-resolution graphs, guiding the selection of nodes to be discarded in the next hierarchy for improved long-range modeling. We now delve into the details of the differentiable node selection method (`DiffSELECT`) for hierarchical graph construction.

In the `DiffSELECT` operation, we train the node update module $\phi^v$ based on anisotropic aggregated edge features to produce a probability $p_i$ for each node. This probability indicates the likelihood of retaining node $v_i$ in the next-level coarser graph $\mathcal{G}_{l+1}$. Accordingly, we rewrite Eq. (4) as follows:

$$\hat{\mathbf{v}}_i^l, p_i^l = \phi^v \Big( \mathbf{v}_i^l, \sum_{v_j \in \mathcal{N}_{v_i}} \alpha_{ij}^l \hat{\mathbf{e}}_{ij}^l \Big). \tag{5}$$

Next, we employ Gumbel-Softmax sampling (Jang et al., 2017) on $p_i$ to determine whether node $v_i$ is included in the downsampled graph:

$$z_i^l = \text{Gumbel-Softmax}\big(p_i^l\big), \tag{6}$$

where $z_i^l$ is a binary variable indicating the selection of node $v_i$. When $z_i^l = 1$, node $v_i$ is retained in the next graph level. In this way, the node set $\mathcal{V}_{l+1}$ is dynamically constructed based on node features from the finer graph level. The Gumbel-Softmax technique provides a differentiable approximation to hard sampling, thereby facilitating end-to-end training. Additionally, we implement the Gumbel-Softmax with temperature annealing to stabilize training, initially encouraging the exploration of hierarchies and gradually refining the selection process.

The edges $\mathcal{E}_{l+1}$ in the coarser graph $\mathcal{G}_{l+1}$ are constructed by connecting the selected nodes using the original graph's edges $\mathcal{E}_l$. However, this process may result in disconnected partitions (see Figure 7 in the appendix). To address this issue, we enhance the connectivity in $\mathcal{E}_{l+1}$ by incorporating the $K$-hop edges during the edge selection process, defined as follows:

$$\widetilde{\mathcal{E}}_l^{(K)} = \mathcal{E}_l \cup \big\{e_{ij} \mid \exists v_{k_1}, v_{k_2}, \ldots, v_{k_{K-1}} \in \mathcal{V}_l \text{ s.t. } e_{i,k_1}, e_{k_1,k_2}, \ldots, e_{k_{K-1},j} \in \mathcal{E}_l\big\}. \tag{7}$$

In essence, $e_{ij} \in \widetilde{\mathcal{E}}_l^K$ if there exists a sequence of intermediate nodes $\{v_{k_1}, v_{k_2}, \ldots, v_{k_{K-1}}\}$ consecutively connected by edges in $\mathcal{E}_l$ or $e_{ij} \in \mathcal{E}_l$. The edges in $\mathcal{E}_{l+1}$ are defined as:

$$\mathcal{E}_{l+1} = \big\{e_{ij} \mid \exists v_i, v_j \in \mathcal{V}_{l+1} \text{ s.t. } e_{ij} \in \widetilde{\mathcal{E}}_l^{(K)}\big\}. \tag{8}$$

$\mathcal{E}_{l+1}$ consists of edges from the enhanced edge set $\widetilde{\mathcal{E}}_l^{(K)}$ that connect nodes in $\mathcal{V}_{l+1}$. As $K$ increases, nodes in $\widetilde{\mathcal{E}}_l^{(K)}$ can be connected through additional intermediate nodes, thereby improving long-range connectivity. In practice, the most effective value of $K$ is 2, which ensures effective connectivity.

The graph construction process is fully differentiable, allowing for seamless integration into differentiable physical simulators. By flexibly adapting graph hierarchies based on simulation states, it paves the way for more accurate predictions of the spatiotemporal patterns in complex systems.

## 3.4 INTER-LEVEL FEATURE PROPAGATION WITH AMP

During the downsampling process from $\mathcal{G}_l$ to the generated coarser graph $\mathcal{G}_{l+1}$, as illustrated in Figure 1, the REDUCE operation aggregates information to each node in $\mathcal{V}_{l+1}$ from its corresponding neighbors in $\mathcal{V}_l$. Conversely, the EXPAND operation unpools the reduced graph back to a finer resolution, delivering the information of the pooled nodes to their neighbors at the finer level.

Prior works employed non-parametric aggregation in inter-level propagation, convolving features based on the normalized node degree. It simplifies intricate relationships between nodes and neglects the directional aspects of information flow. In comparison, the inter-level aggregation weights in DHMP are data-specific and time-varying. Notably, the importance weight $\alpha_{ij}^l$ in the proposed AMP layer inherently captures the significance of node $v_j$'s features to node $v_i$ at the graph level $l$. Consequently, it can be directly reused for the REDUCE and EXPAND operations in the downsampling and upsampling processes. We provide details of these operations as follows:

- REDUCE: Let $v_i$ be the node at the coarser graph level. The downsampling process aggregates the information of the current neighbors $\mathcal{N}_i$ by reusing the weight $\alpha_{ij}^l$: $\mathbf{v}_i^{l+1} \leftarrow \sum_{j \in \mathcal{N}_i} \alpha_{ij}^l \mathbf{v}_j^l$.

- EXPAND: We first unpool the node features from the next-level coarser graph $\mathcal{G}_{l+1}$. To achieve this, we record the nodes selected during the downsampling process and use this information to place nodes back in their original positions in the graph. Next, we re-use the importance weight $\alpha_{ij}^l$ to assign features in the coarser graph to nodes in the finer graph, *i.e.*, $\mathbf{v}_i^l \leftarrow \sum_{j \in \mathcal{N}_i} \mathbf{v}_j^{l+1} \alpha_{ij}^l$.

- FeatureMixing: Following the EXPAND operation, DHMP conducts an additional message passing step based on $\mathbf{v}_i^l$. It then integrates the resulting features with the intra-level message passing outcomes in $\mathcal{G}_l$ (before downsampling) through a skip connection.

## 3.5 IMPLEMENTATION DETAILS

We train DHMP using the one-step supervision that measures the $L_2$ loss between the ground truth and the next-step predictions. We include detailed descriptions of the physical quantities represented

by input node and edge features in Appendix B. We implement the encoder, decoder, node update function $\phi^v$, and edge update function $\phi^e$ using two-layer MLPs with ReLU activation and a hidden size of 128. Likewise, the network component for generating importance weights, $\phi^w$, in AMP is implemented using a two-layer MLP. We apply layer normalization to the MLP outputs, except for those of the decoder and the importance weight network. We perform a single message passing step at each graph level. We discuss the specific number of graph levels $L$ for downsampling in Appendix A. In the Gumbel-Softmax operator for differentiable node selection, we use temperature annealing to decrease the temperature from 5 to 0.1 with a decay factor of $\gamma = 0.999$, which aims to encourage the exploration of hierarchies while gradually refining node selection to ensure training stability.

## 4 EXPERIMENTS

### 4.1 EXPERIMENTAL SETUP

**Datasets.** We evaluate our approach on five mesh-based physics simulation benchmarks established in previous literature (Pfaff et al., 2021; Cao et al., 2023; Wu et al., 2023; Narain et al., 2012).

- *CylinderFlow*: Simulation of incompressible flow around a cylinder based on 2D Eulerian meshes.
- *Airfoil*: Aerodynamic simulation around airfoil cross-sections based on 2D Eulerian meshes.
- *Flag*: Simulation of flag dynamics in the wind based on Lagrangian meshes with fixed topology.
- *DeformingPlate*: Deformation of hyper-elastic plates based on Lagrange tetrahedral meshes.
- *FoldingPaper*: Deformation of paper sheets on Lagrangian meshes, with varying forces at the four corners and evolving mesh graph.

For details regarding the datasets, including descriptions of the input physical quantities, please refer to Appendix A. Additional information concerning our implementation can be found in Appendix B.

**Compared models.** We primarily compare DHMP with the following methods:

- MGN (Pfaff et al., 2021), which performs multiple times of message passing at the original graph.
- BSMS-GNN (Cao et al., 2023), which generates static hierarchies using bi-stride pooling and performs message passing on predefined meshes.
- Lino et al. (2022), which also trains MPNNs on manually-set multi-scale mesh graphs.
- HCMT (Yu et al., 2024), which generates static hierarchies by applying Delaunay triangulation to the bi-stride pooled nodes, and enables directed feature propagation with the attention mechanism.

All models are trained using the Adam optimizer with an exponential learning rate decay from $10^{-4}$ to $10^{-6}$. We further clarify the architecture details and the hyperparameters in Appendix C.

### 4.2 MAIN RESULTS

**Standard benchmarks.** Table 2 presents the root mean squared error (RMSE) of one-step prediction (RMSE-1) and long-term rollouts for 100–600 future time steps (RMSE-all). DHMP consistently outperforms the compared models across all benchmarks. This demonstrates the effectiveness of building context-aware, temporally evolving hierarchies with learnable, directionally non-uniform

Table 2: Quantitative comparison of the one-step and long-term prediction errors. We report the mean results over 3 random seeds, with corresponding standard deviations detailed in Appendix F. *Promotion* denotes the improvement over the second-best model.

| Model | RMSE-1 ($\times 10^{-2}$) | | | | RMSE-All ($\times 10^{-2}$) | | | |
|---|---|---|---|---|---|---|---|---|
| | Cylinder | Airfoil | Flag | Plate | Cylinder | Airfoil | Flag | Plate |
| MGN (2021) | 0.4046 | 77.38 | 0.4890 | 0.0579 | 59.78 | 2816 | 124.5 | 3.982 |
| BSMS-GNN (2023) | 0.2263 | 71.69 | 0.5080 | 0.0632 | 16.98 | 2493 | 168.1 | 1.811 |
| Lino et al. (2022) | 3.9352 | 85.66 | 0.9993 | 0.0291 | 27.60 | 2080 | 118.2 | 2.090 |
| HCMT (2024) | 0.9190 | 48.62 | 0.4013 | 0.0295 | 23.59 | 3238 | 90.32 | 2.468 |
| DHMP | **0.1568** | **41.41** | **0.3049** | **0.0282** | **6.571** | **2002** | **76.16** | **1.296** |
| *Promotion* | **30.7%** | **14.8%** | **24.0%** | **3.10%** | **61.3%** | **3.75%** | **15.7%** | **28.5%** |

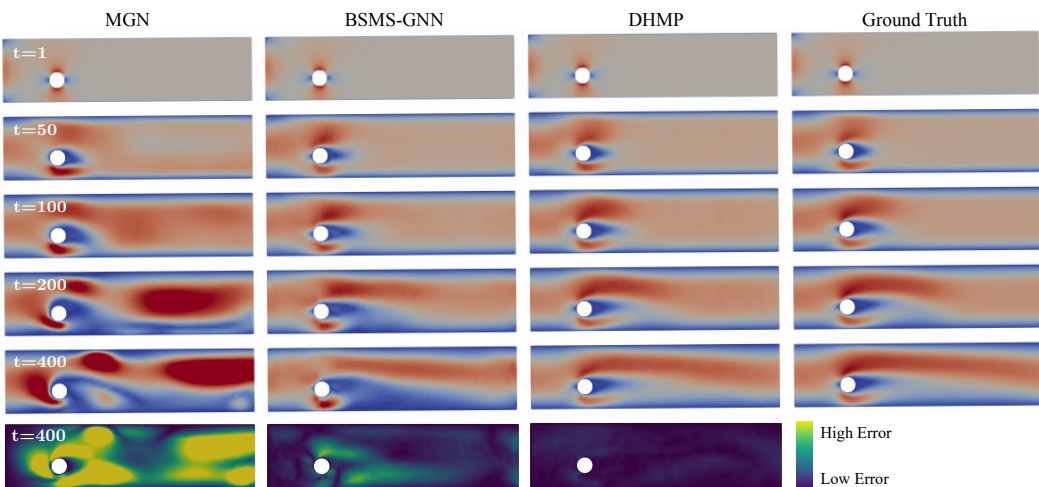

Figure 2: Prediction showcases over 400 future steps on CylinderFlow and the final error maps.

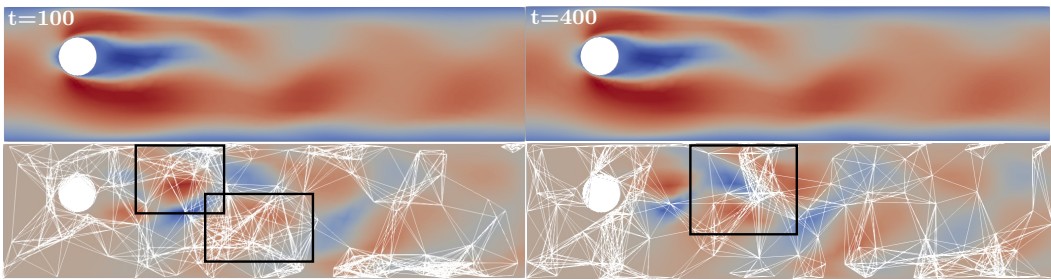

Figure 3: **Top:** the velocity field from the true data. **Bottom:** the temporal difference of the velocity fields between adjacent time steps alongside the constructed coarser-level mesh graph ($\mathcal{G}_{l=4}$). The highlighted areas demonstrate a notable experimental phenomenon: the mesh dynamically evolves with the data context, and aligns with the critical areas of change in the data.

feature propagation both within and across graph levels. Figure 2 presents long-term predictions on CylinderFlow, based solely on the system's initial conditions at the first step. As we can see, DHMP captures the complex, time-varying fluid flow around the cylinder obstacle more successfully, with its predictions closely matching the ground truth evolution. More results are shown in Appendix I.

**Can the learned hierarchies adapt to evolving data dynamics?** In Figure 3, we visualize the dynamic hierarchies constructed by DHMP at different time steps, where coarser-level nodes tend to concentrate in regions highlighted by the temporal differences in the true data. We have two observations here: First, the constructed hierarchy evolves as the data context changes. Second, the temporally evolving graph structures align with the high-intensity regions, either in the velocity fields (top) or in their temporal variations (bottom). These findings highlight the effectiveness of our approach in capturing significant patterns within the dynamic system.

**Paper simulation with changing meshes.** We evaluate DHMP in a more challenging setting with time-varying meshes for paper folding simulation, generated using the ARCSim solver (Narain et al., 2012; Wu et al., 2023). Notably, methods such as BSMS-GNN and HCMT require pre-computed hierarchies as part of their data preprocessing, which poses a significant limitation in scenarios with

Table 3: Errors of 2D paper simulation ($\times 10^{-2}$) on time-varying meshes.

| Model | RMSE-1 | RMSE-All |
|-------|--------|----------|
| MGN | 0.0618 | 24.08 |
| DHMP | **0.0544** | **7.41** |

continuously changing mesh topologies. We assess the models using ground-truth remeshing nodes provided by the ARCSim Adaptive Remeshing component, following the setup from Pfaff et al. (2021). As shown in Table 3, DHMP achieves superior short-term and long-term accuracy compared to MGN, indicating that the dynamic graph hierarchies in our approach can better fit physical systems with significant geometric variations, as represented by the time-varying input mesh structures.

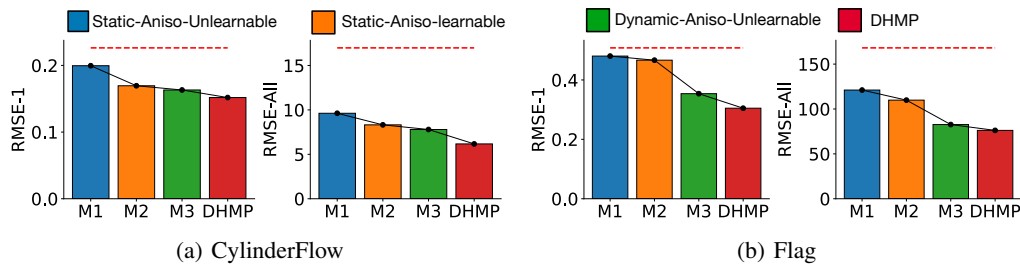

(a) CylinderFlow      (b) Flag

Figure 4: The analyses of dynamic hierarchies, anisotropic intra-level propagation, and learnable inter-level feature propagation. The red dashed lines represent results from BSMS-GNN (Cao et al., 2023). Lower values indicate better performance.

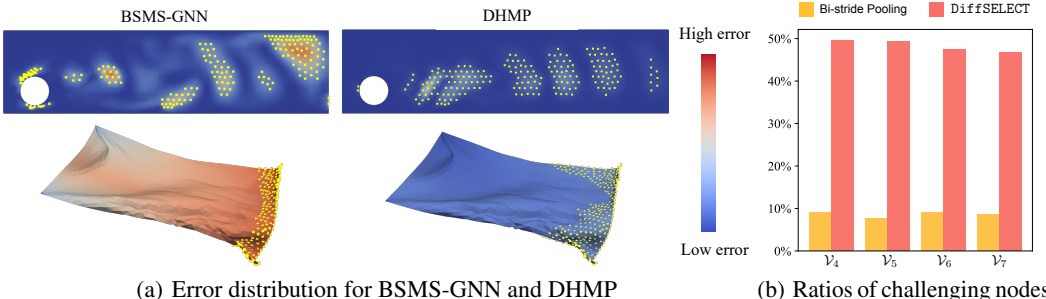

(a) Error distribution for BSMS-GNN and DHMP      (b) Ratios of challenging nodes

Figure 5: (a) Error maps, where nodes with the top $10\%$ of errors in each model's predictions are marked in yellow and referred to as "*challenging nodes*". (b) DHMP retains more challenging nodes in coarser graph hierarchies to capture multi-scale dependencies more effectively.

**Model stability under variable graph structures.** Due to the stochasticity of Gumbel-Softmax sampling in `DiffSELECT`, we evaluate the stability of trained DHMP by conducting three independent runs on the test set. The mean and standard deviations of the prediction errors reveal minimal discrepancies across different runs, as shown in Table 11 in Appendix E. These findings demonstrate that once trained, DHMP generates consistent graph hierarchies based on the same inputs.

**Computation efficiency.** The computation efficiency is evaluated in Appendix G. It shows that DHMP has the lowest training cost and lower inference time compared to attention-based model.

## 4.3 ABLATION STUDIES

DHMP has three contributions: *(i)* dynamic hierarchy, *(ii)* anisotropic intra-level propagation, *(iii)* learnable inter-level propagation. To investigate the effectiveness of each component, we implement various variants of DHMP, including *Static-Anisotropic-Unlearnable (M1)*, *Static-Anisotropic-Learnable (M2)*; *Dynamic-Anisotropic-Unlearnable (M3)*, and compare them against BSMS-GNN. The baseline model uses static hierarchies, isotropic intra-level summation, and unlearnable inter-level propagation. Ablation study results compared to baseline BSMS-GNN are presented in Figure 4.

**Effectiveness of dynamic hierarchies.** From Figure 4, by comparing DHMP vs. *M2* and *M3* vs. *M1*, we observe the advantages of learning dynamic hierarchical graph structures. These results highlight the significance of adaptively modeling interactions in context-dependent graphs. To better understand how DHMP constructs dynamic hierarchies, we visualize the distribution of nodes with the top $10\%$ prediction errors in Figure 5(a). Accordingly in Figure 5(b), we observe that DHMP retains a higher proportion of "challenging" nodes in the coarser message passing levels, enabling our model to capture multi-scale dependencies more effectively, especially in areas where finer message passing levels struggle. In contrast, the predefined static hierarchies in the BSMS baseline are data-independent and may inevitably overlook modeling long-range relations surrounding these pivotal nodes, even though they typically present higher errors than those in DHMP.

**Effectiveness of anisotropic message passing.** Figure 4 further illustrates the importance of enhancing the direction-specific contributions during both intra-level and inter-level updates. First, incorporating AMP into the static hierarchy results in performance improvements, as shown by the comparison between *M1* and BSMS-GNN. Additionally, the significance of transmitting directed

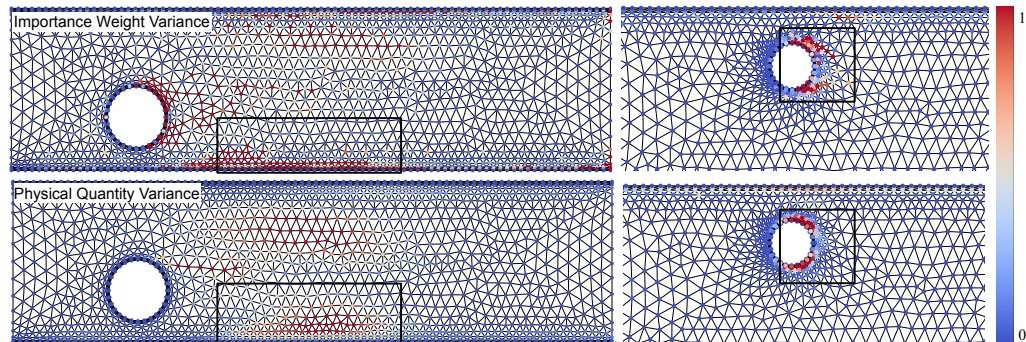

Figure 6: Visualizations of the variance of the generated anisotropic weights calculated on adjacent edges (**top**) and the corresponding variance of physical quantities computed over time (**bottom**). The strong correlation between them reveals the AMP's ability to perceive significant patterns in data.

Table 4: Results on out-of-distribution (OOD) mesh resolutions.

| Model | RMSE-1 ($\times 10^{-2}$) | | | | RMSE-All ($\times 10^{-2}$) | | | |
|---|---|---|---|---|---|---|---|---|
| | Cylinder | Airfoil | Flag | Plate | Cylinder | Airfoil | Flag | Plate |
| BSMS-GNN | 0.9177 | 202.3 | 0.6486 | 0.0474 | **33.87** | 6179 | 148.2 | **1.904** |
| DHMP | **0.4855** | **126.7** | **0.5536** | **0.0368** | 47.72 | **5759** | **120.9** | 2.553 |

inter-level information is highlighted by comparing DHMP vs. *M3* and *M1* vs. *M2*. In Figure 6, we visualize the variance of predicted anisotropic edge weights and compare it with areas where physical quantities present substantial variations over time. The results reveal a strong correlation between the anisotropic learning mechanism and the rapidly changing dynamics of the physical system.

### 4.4 GENERALIZATION ANALYSES

**Generalization to out-of-distribution mesh resolutions.** Almost none of the existing machine learning models for mesh-based physics simulation are resolution-free. They may fail when evaluated on unseen mesh resolutions. We assess the generalization performance of DHMP by training it on low-resolution meshes and testing it on high-resolution meshes. The average number of nodes in the test data is twice that of the training data, and the number of edges is three times greater. As shown in Table 4, DHMP demonstrates improved zero-shot generalization ability to more refined mesh structures. This improvement is primarily attributed to our model's capability to generate hierarchical graphs adaptively. However, it is important to note that this result does not imply that our method has fully explored generalization across arbitrary resolutions—achieving true resolution-free modeling requires a more refined model design. Nevertheless, this holds significant value in practical applications and has the potential to greatly reduce the time overhead of numerical simulation processes for preparing the large-scale mesh data required for model training.

**Generalization to physics variations.** We evaluate DHMP under strong distribution shifts in the input physical quantities. Table 5 presents data statistics and the RMSE results on the CylinderFlow and Airfoil datasets. DHMP achieves lower RMSEs than BSMS-GNN in both short-term and long-term simulations, which can be largely attributed to the proposed AMP layer. When the fluid dynamics in the test set become more complex—characterized by increased variance in the velocity field over time—the dynamics patterns propagate more rapidly in space. The AMP layer can more effectively capture directed long-range node interactions.

Table 5: Generalization results across various scales of input velocities, presented by the variance and norm of data in training/test splits. *Increase* denotes the relative increase of the test data compared to the training data.

| | Cylinder | | Airfoil | |
|---|---|---|---|---|
| Split | Var | Norm | Var | Norm |
| Train | 7.92 | 579.6 | 288.3 | 173.4 |
| Test | 13.43 | 826.3 | 827.4 | 180.6 |
| *Increase* | 64.5% | 42.5% | 186.9% | 4.20% |
| Model | RMSE-1 | RMSE-All | RMSE-1 | RMSE-All |
| BSMS-GNN | $2.58 \times 10^{-3}$ | 0.251 | 1.035 | 30.32 |
| DHMP | $\mathbf{2.14 \times 10^{-3}}$ | **0.091** | **0.665** | **22.57** |

## 5 RELATED WORK

**Learning-based physics simulation.**    Recent literature has shown that learning-based simulators can efficiently handle complex and high-dimensional problems, such as fluid dynamics (Zhu et al., 2024), structural analysis (Kavvas et al., 2018; Thai, 2022), and climate modeling (Kurth et al., 2018; Rasp et al., 2018; Rolnick et al., 2022; Lam et al., 2023). The models can be roughly categorized into three groups based on data representation: those modeling partial differential equations (Raissi et al., 2017; 2019; Lu et al., 2019; Li et al., 2021; Wang et al., 2021), particle-based systems (Li et al., 2019; Sanchez-Gonzalez et al., 2020; Ummenhofer et al., 2020; Prantl et al., 2022), and mesh-based systems (Pfaff et al., 2021; Lino et al., 2022; Fortunato et al., 2022; Cao et al., 2023). The rapid inference time and differentiable property of these models greatly facilitate downstream tasks, such as inverse design (Wang & Zhang, 2021; Goodrich et al., 2021; Allen et al., 2022; Janny et al., 2023).

**GNN-based physics simulation.**    Previous work has explored GNNs in various physical domains, such as articulated systems (Sanchez-Gonzalez et al., 2018), soft-body deformation and fluids (Li et al., 2019; Mrowca et al., 2018; Sanchez-Gonzalez et al., 2020; Rubanova et al., 2022; Wu et al., 2023), rigid body dynamics (Battaglia et al., 2016; Li et al., 2019; Mrowca et al., 2018; Bear et al., 2021; Rubanova et al., 2022), and aerodynamics (Belbute-Peres et al., 2020; Hines & Bekemeyer, 2023; Pfaff et al., 2021; Fortunato et al., 2022; Cao et al., 2023). Among them, MGN (Pfaff et al., 2021) is a key method that models mesh-based dynamics through graph interactions. Subsequent approaches primarily focus on enhancing modeling capabilities and reducing computational costs.

**Hierarchical GNNs for physics simulation.**    Hierarchical GNNs employ multi-scale graph structures (Lino et al., 2022; Han et al., 2022; Fortunato et al., 2022; Allen et al., 2023; Janny et al., 2023; Cao et al., 2023; Yu et al., 2024) to decrease overhead by using fewer nodes at coarser levels and enabling long-range feature propagation. GMR-Transformer-GMUS (Han et al., 2022) employs a uniform sampling pooling method to select pivotal nodes. MS-MGN (Fortunato et al., 2022) uses a dual-level hierarchical GNN and performs message passing at both fine and coarse resolutions. Hierarchical GNNs with multi-level structures (Lino et al., 2022; Cao et al., 2023; Yu et al., 2024; Garnier et al., 2024; Hy & Kondor, 2023) are most relevant to our approach, as they integrate message passing neural networks within the U-Net architecture (Ronneberger et al., 2015). Lino et al. (2022) uses manually set grid resolutions and spatial proximity for graph pooling, which requires predefined parameters. BSMS-GNN (Cao et al., 2023) introduces a bi-stride pooling strategy that pools nodes on alternating breadth-first search frontiers while enhancing edges with two-hop connections. HCMT (Yu et al., 2024) refines the structure further by applying Delaunay triangulation to bi-stride nodes. Notably, these methods construct multi-level structures as preprocessing and cannot change the graph hierarchies under varying physical conditions. Moreover, they typically use uniform feature aggregation for intra-level propagation, which may hinder the directed transfer of significant dynamic patterns, or use attention-based aggregation, which increases computational overhead. Furthermore, inter-level propagation is often predefined or unlearnable, limiting flexibility in transferring information across hierarchy levels. In contrast, our model generates context-aware and temporally evolving graph hierarchies and incorporates learnable anisotropic feature propagation, allowing it to better adapt to various initial conditions and rapidly changing dynamic systems.

## 6 CONCLUSIONS AND LIMITATIONS

In this paper, we introduced DHMP, a neural network that significantly advances the state-of-the-art in mesh-based physics simulation. Our key innovation is dynamically creating the context-aware graph structures of hierarchical GNNs through a differentiable node selection process. To this end, we proposed an anisotropic message passing mechanism to enhance the propagation of long-term dependencies between distant nodes, aligning with the directed nature of significant dynamic patterns. Extensive experiments show that DHMP outperforms existing models, especially those with fixed graph hierarchies, in both short-term and long-term predictions.

A potential limitation of this work is the need to improve the interpretability of the learned hierarchy structure. Additionally, we would consider incorporating specific physical priors into DHMP to further enhance the model's robustness and generalizability, particularly in *resolution-free* problem settings, which have been less explored in existing mesh-based approaches.

## ETHICS STATEMENT

In this work, we adhere to the highest ethical standards across all stages of research. No human subjects were involved, and no personal data was used, ensuring compliance with privacy and security protocols. All datasets utilized are publicly available, mitigating concerns related to sensitive information exposure. We acknowledge the potential use of physics simulation models for harmful insights if misapplied; therefore, we encourage careful consideration of the context and application domain when deploying these models.

## REPRODUCIBILITY STATEMENT

We mainly build DHMP based on the released code of BSMS-GNN. We prioritize the repeatability of our work and will open source the source code. All results can be reproduced by following the experimental details presented in Section 4.1 and Appendices A–C.

### ACKNOWLEDGMENTS

This work was supported by the National Natural Science Foundation of China (Grant No. 62250062, 62106144), the Shanghai Municipal Science and Technology Major Project (Grant No. 2021SHZDZX0102), and the Fundamental Research Funds for the Central Universities.

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

APPENDIX

## A DATASETS

We employ four established datasets from MGN (Pfaff et al., 2021): *CylinderFlow*, *Airfoil*, *Flag*, and *DeformingPlate*.

- The CylinderFlow case examines the transient incompressible flow field around a fixed cylinder positioned at different locations, with varying inflow velocities.
- The Airfoil case explores the transient compressible flow field at varying Mach numbers around the airfoil, with different angles of attack.
- The Flag case involves a flag blowing in the wind on a fixed Lagrangian mesh.
- The DeformingPlate case involves hyperelastic plates being compressed by moving obstacles.

The CylinderFlow, Airfoil, and Flag datasets are each split into 1,000 training sequences, 100 validation sequences, and 100 testing sequences. The DeformingPlate dataset is split into 500 training sequences, 100 validation sequences, and 100 testing sequences.

We also consider a more challenging dataset, *FoldingPaper*, where varying forces at the four corners deform paper with time-varying Lagrangian mesh graphs, generated using the ARCSim solver (Narain et al., 2012; Wu et al., 2023). This dataset is divided into 500 training sequences, 100 validation sequences, and 100 testing sequences.

We present statistical details of all five datasets in Table 6 and the input physical quantities in Table 7.

Table 6: Statistics of the CylinderFlow, Airfoil, Flag, DeformingPlate, and FoldingPaper datasets.

| Dataset | Average # nodes | Average # edges | Mesh type | # Hierarchies | # Steps |
|---|---|---|---|---|---|
| CylinderFlow | 1886 | 5424 | triangle, 2D | 7 | 600 |
| Airfoil | 5233 | 15449 | triangle, 2D | 7 | 100 |
| Flag | 1579 | 9212 | triangle, 2D | 7 | 400 |
| DeformingPlate | 1271 | 4611 | tetrahedron, 3D | 6 | 400 |
| FoldingPaper | 110 | 724 | triangle, 2D | 3 | 325 |

Table 7: Comparisons of the edge offsets and node inputs of different physical systems.

| Dataset | Type | Edge offset $\mathbf{e}_{ij}$ | Node Input $\mathbf{v}_i$ | Outputs | Noise Scale |
|---|---|---|---|---|---|
| CylinderFlow | Eulerian | $X_{ij}, \lvert X_{ij} \rvert$ | $v_i, n_i$ | $\dot{v}_i$ | $v_i : 2e-2$ |
| Airfoil | Eulerian | $X_{ij}, \lvert X_{ij} \rvert$ | $\rho_i, v_i, n_i$ | $\dot{v}_i, \dot{\rho}_i, P_i$ | $v_i : 2e-2, \rho_i : 1e1$ |
| Flag | Lagrangian | $X_{ij}, \lvert X_{ij} \rvert, x_{ij}, \lvert x_{ij} \rvert$ | $\dot{x}_i, n_i$ | $\dot{x}_i$ | $x_i : 3e-3$ |
| DeformingPlate | Lagrangian | $X_{ij}, \lvert X_{ij} \rvert, x_{ij}, \lvert x_{ij} \rvert$ | $\dot{x}_i, n_i$ | $\dot{x}_i$ | $x_i : 3e-3$ |

## B MODEL IMPLEMENTATION

We present model configurations of different physical systems below:

- **Edge offsets.** $X$ and $x$ stand for the mesh-space and world-space position. For an Eulerian system, only mesh position is used for $\mathbf{e}_{ij}$, while for a Lagrangian system, both mesh-space and world-space positions are used. The edge offsets are directly used as low-dimensional input to the edge update function $\phi^e$. In other words, these features are concatenated and fed into $\phi^e$ without any transformation through an MLP or other encoding processes to generate a higher-dimensional representation.

- **Input and target of the physical term of node** $v_i$. $v$ is the velocity, $\rho$ is the density, $P$ is the absolute pressure, and the dot $\dot{a} = a_{t+1} - a_t$ stands for temporal change for a variable $a$. $n$ stands for the node type of $v_i$. Random Gaussian noise is added to the node input features to enhance robustness during training (Pfaff et al., 2021; Sanchez-Gonzalez et al., 2020; Cao et al., 2023). All the variables involved are normalized to zero-mean and unit variance via preprocessing.

The preprocessed physical term is fed to the encoder to transform it into a high-dimensional representation.

The encoder, decoder, node update function $\phi^v$, and edge update function $\phi^e$ all utilize two-layer MLPs with ReLU activation and a hidden size of 128. Similarly, the importance weight network $\phi^w$ in AMP is implemented using a two-layer MLP. LayerNorm is applied to the MLP outputs, except for the decoder and the importance weight network. We set $K = 2$ for edge enhancement, which is aligned with the setting of BSMS-GNN (Cao et al., 2023). In the Gumbel-Softmax for differentiable node selection, temperature annealing decreases the temperature from 5 to 0.1 using a decay factor of $\gamma = 0.999$, encouraging exploration of hierarchies while gradually refining their selection to ensure stability. DHMP is trained with Adam optimizer, using an exponential learning rate decay from $10^{-4}$ to $10^{-6}$. All experiments are conducted using 4 Nvidia RTX 3090. We mainly build DHMP based on the released code of BSMS-GNN (Cao et al., 2023).

## C  BASELINE DETAILS

We compare DHMP with four competitive baselines: (1) MGN (Pfaff et al., 2021) which performs multiple message passing on the input high-resolution mesh topology; (2) BSMS-GNN (Cao et al., 2023), which uses predefined bi-stride pooling prior as preprocessing to generate static hierarchies on same mesh topology; (3) Lino *et al.*(Lino et al., 2022), which uses manually set grid resolutions and spatial proximity for graph pooling; (4) HCMT (Yu et al., 2024), which uses Delauny triangulation based on bi-stride nodes and adopt attention mechanism to enable non-uniform feature propagation. The architecture details of the compared models are as follows:

- **MGN.**  In MGN, we use 15 message passing steps in all datasets. The encoder, decoder, node update function, and edge update function are configured in the same way as in our model.

- **BSMS-GNN.**  We use the same number of graph hierarchies in DHMP and as in BSMS-GNN. We use the minimum average distance as the seeding heuristic for the BFS search recommended in its original paper. The multi-level building is processed in one pass. The inter-level propagation uses the normalized node degree to convolve features from neighbors to central nodes. The encoder, decoder, node update function, and edge update function are set up the same way as in our model. We perform one message passing step at each graph level.

- **Lino *et al.***  We use the four-scale GNN structure proposed in the work of Lino et al. (2022). The edge length of the smallest cell for each dataset is $1/10$ of the average scene size, with each lower scale doubling in size. We follow its original paper to use 4 message passing steps at the top and bottom levels and two for the others.

- **HCMT.**  The hidden dimension and the number of attention heads in the HCMT block are set to 128 and 4, respectively. We use the same number of hierarchies as in DHMP. For the Cylinder and Airfoil datasets, due to the presence of hollow sections in the mesh, we do not apply Delaunay triangulation for remeshing. Instead, we use edge connections generated through bi-stride pooling. Like in DHMP, we use a single message passing step at each graph level.

Notably, the node encoder, decoder, node update function, and edge update function of MGN, BSMS-GNN, and Lino *et al.* have the same network architecture as those in DHMP. To reduce the number of network parameters, we avoid separately encoding the edge offset $\mathbf{e}_{ij}$. Instead, we concatenate it with the node latents and use this combined input for the edge update function to compute $\hat{\mathbf{e}}_{\mathbf{ij}}$.

All models are trained using the Adam optimizer, with an exponential learning rate decay from $10^{-4}$ to $10^{-6}$ and a decay rate of $\gamma = 0.79$. The batch size is set to 32. Following BSMS-GNN, model convergence is defined by a performance improvement threshold of $< 1\%$, at which point the training process is terminated.

## D  ADDITIONAL RESULTS

### D.1  ABLATION STUDY

In Sec. 4.3, we compare different variants of our DHMP model against the BSMS-GNN baseline, to evaluate the effectiveness of *(i)* dynamic hierarchy construction based on the input mesh topology and

Table 8: Qualitative results of model variants of DHMP and the baseline model.

| Model | RMSE-1 ($\times 10^{-2}$) | | RMSE-All ($\times 10^{-2}$) | |
| --- | --- | --- | --- | --- |
| | Cylinder | Flag | Cylinder | Flag |
| BSMS-GNN (Cao et al., 2023) | 0.2263 | 0.5080 | 16.98 | 168.1 |
| Static-Anisotropic-Unlearnable (M1) | 0.1995 | 0.4804 | 9.621 | 121.1 |
| Static-Anisotropic-Learnable (M2) | 0.1695 | 0.4666 | 8.317 | 109.9 |
| Dynamic-Anisotropic-Unlearnable (M3) | 0.1631 | 0.3538 | 7.793 | 82.65 |
| DHMP | **0.1568** | **0.3049** | **6.571** | **76.16** |

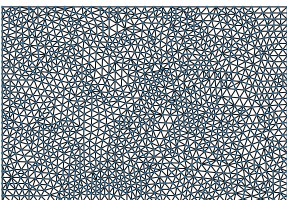 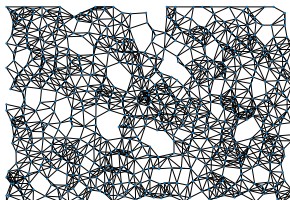 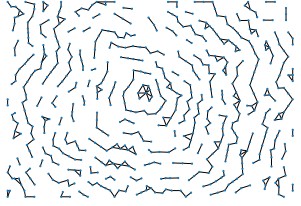

Figure 7: Mesh visualization on Flag Dataset. Original mesh (*left*), sub-level graph after differentiable node selection with $K$-hop enhancement with $K = 2$ (*middle*), and sub-level graph after node selection without $K$-hop enhancement (*right*).

physical quantities, *(ii)* anisotropic intra-level feature propagation, *(iii)* learnable inter-level feature propagation. The variants we investigate include:

- *Static-Anisotropic-Unlearnable (M1): (ii)*,
- *Static-Anisotropic-Learnable (M2): (ii+ iii)*,
- *Dynamic-Anisotropic-Unlearnable (M3): (i)+(ii)*.

In this ablation study, we utilize a static graph hierarchy preprocessed using bi-stride pooling as described in the BSMS-GNN paper (Cao et al., 2023), along with a non-parametric intra-level aggregation function from previous works (Pfaff et al., 2021; Cao et al., 2023). Additionally, BSMS-GNN employs unlearnable node degree metrics to generate inter-level aggregation weights, which convolve features based on the normalized node degree for inter-level propagation. We show the quantitative RMSE values of Figure 4 in Table 8.

## D.2 EDGE ENHANCEMENT

When constructing the lower-level graph $\mathcal{G}_{l+1}$ based on the selected nodes, the edges $\mathcal{E}_{l+1}$ are formed by connecting these nodes using the original edges $\mathcal{E}_l$ from the previous graph. However, this approach may lead to disconnected partitions, as observed in previous work (Lee et al., 2019; Cao et al., 2023; Gao & Ji, 2019), and illustrated in Figure 7. To address this issue, we enhance the connectivity of $\mathcal{E}_{l+1}$ by incorporating $K$-hop edges during the edge construction process. We investigate the impact of different $K$ values, specifically $K = 2, 3, 4$, on the Flag dataset. The results are presented in Table 9, along with comparisons of the computational efficiency.

Notably, $K = 2$ yields the lowest RMSE across all conditions (RMSE-1, RMSE-50, and RMSE-all), indicating superior performance compared to higher $K$ values. Despite the performance decline observed with $K = 3$ and $K = 4$, they still outperform the baseline results, indicating the effectiveness of dynamic hierarchical modeling and anisotropy message passing.

## D.3 IMPACT OF EDGE OFFSET ENCODING

To align with the original implementation of MGN (Pfaff et al., 2021), we conduct additional experiments on the CylinderFlow dataset where we implement MGN, BSMS-GNN, and DHMP with *edge offset encoding*. The results are illustrated in Figure 8, where we have the following observations.

Table 9: Results for different values of $K$ in edge enhancement. Here, $K = 1$ denotes directly using edges of selected nodes from previous graph levels. Training time and memory usage are measured with a batch size of 32, while inference time and memory are evaluated with a batch size of 1.

|  | RMSE-1 ($\times 10^{-2}$) | RMSE-All ($\times 10^{-2}$) | Training | | Infer | |
| --- | --- | --- | --- | --- | --- | --- |
|  |  |  | Time (ms) | vRAM (GBs) | Time (ms) | vRAM (GBs) |
| $K = 1$ | 0.3296 | 100.1 | **31.57** | **14.75** | **23.60** | **1.17** |
| $K = 2$ | **0.3049** | **76.16** | 33.67 | 16.53 | 26.33 | 1.24 |
| $K = 3$ | 0.3380 | 86.84 | 34.67 | 18.49 | 33.21 | 1.25 |
| $K = 4$ | 0.3510 | 105.4 | 35.27 | 18.76 | 32.25 | 1.28 |

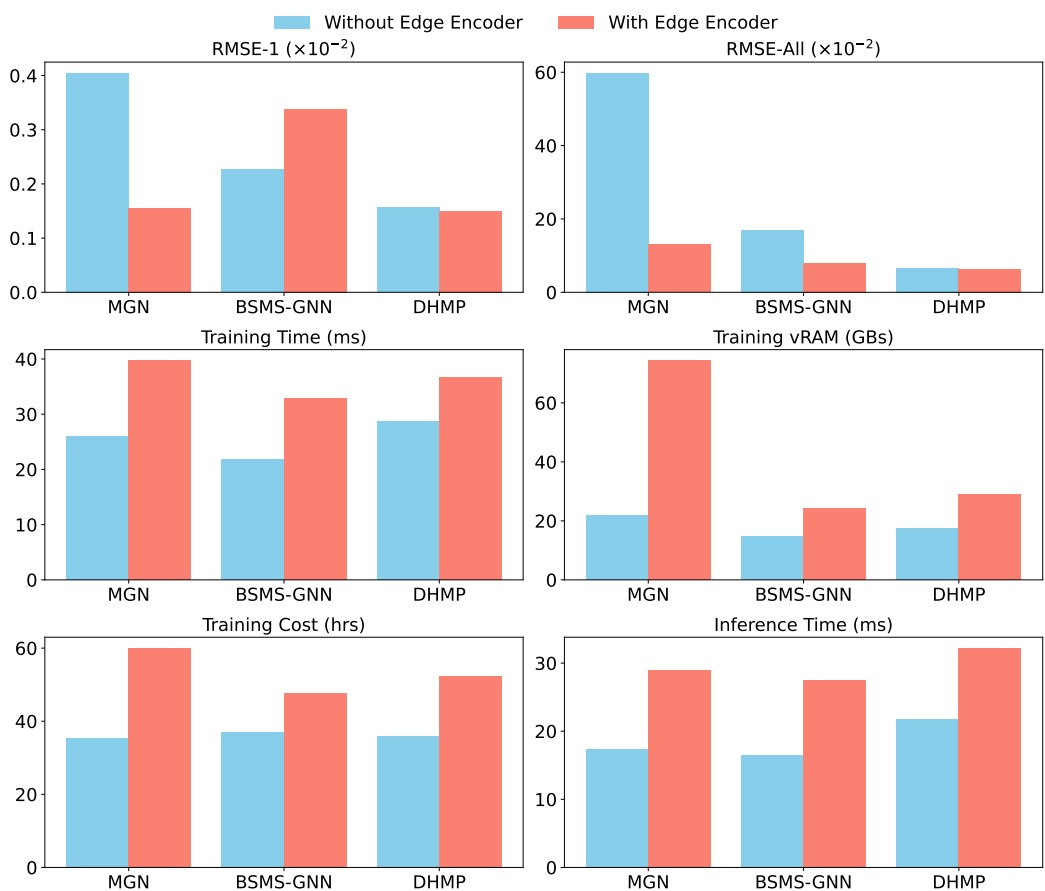

Figure 8: Model comparisons *without vs. with* edge encoding in the CylinderFlow dataset.

First, while edge encoding generally improves model accuracy, it introduces a significant computational overhead. For example, MGN with edge encoding results in a 3x increase in vRAM usage and longer training times compared to the version without edge encoding. For larger datasets, such as Airfoil (which has three times the number of nodes and edges as CylinderFlow), this overhead is expected to be even more pronounced. The increased demands on memory and processing time make it challenging to run these computations on limited GPU resources.

Second, DHMP (with or without edge encoding) consistently outperforms other models in terms of both RMSE-1 and RMSE-All, even when compared to models with edge encoding. Therefore, in our main manuscript, we compare all models using versions without edge encoding to mitigate the substantial increase in computational requirements. We believe this approach provides a fair comparison.

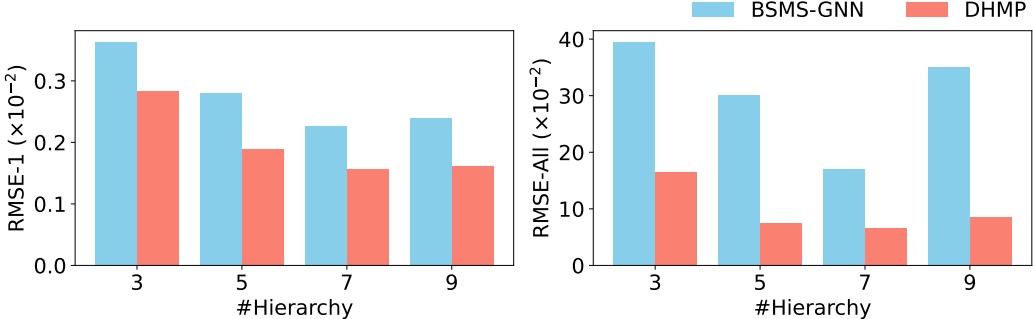

Figure 9: Model comparisons on different numbers of graph hierarchies.

Table 10: Comparison of DHMP *without vs. with KP-AMP* on the CylinderFlow dataset.

|  | RMSE-1 ($\times 10^{-2}$) | RMSE-All ($\times 10^{-2}$) |
| --- | --- | --- |
| DHMP | 0.1568 | 6.571 |
| DHMP with KP-AMP | **0.1438** | **6.444** |

## D.4 HYPERPARAMETER ANALYSES ON NUMBER OF HIERARCHIES

We conduct an ablation study to assess the impact of varying numbers of hierarchies on model performance. The results from the CylinderFlow dataset, illustrated in Figure 9, demonstrate that DHMP consistently outperforms BSMS-GNN across all tested numbers of graph hierarchies. Both models show improved performance with increased hierarchy depth up to 7, indicating that deeper levels help capture more complex interactions and thus enhance accuracy. However, a slight performance decline is observed at level 9, which may suggest the onset of overfitting. Overall, the dynamically learned hierarchies in DHMP are shown to be more effective compared to the predefined static hierarchies used in BSMS-GNN.

## D.5 FURTHER IMPROVEMENT WITH KP-AMP

Recent work by Feng et al. (2022) introduces a novel framework for graph neural networks, emphasizing the distinct processing of information from different hop distances within a graph. In their approach, each hop distance is treated as a separate entity, with dedicated MLPs used to process the messages passing through edges of different hop lengths. This design enables the model to learn varying structural features at different scales, enhancing its expressiveness and adaptability to heterogeneous graph structures.

Inspired by this approach, we explore the applicability of a similar approach to enhance DHMP. Specifically, we extend DHMP by introducing a "KP-AMP" block, characterized by the following modifications:

• The original AMP block is replaced with a specialized KP-AMP block.

• Edges at each hop distance are segregated into separate sets, enabling each hop to be processed independently through a dedicated MLP.

We conduct experiments on the CylinderFlow dataset and showcase the performance comparison between the original DHMP and "DHMP with KP-AMP" in Table 10. The results indicate that incorporating distinct MLPs for each hop distance significantly enhances the model's capability to process structural information at different scales, leading to improved performance across both RMSE-1 and RMSE-All metrics. This approach complements DHMP's anisotropic weighting mechanism by further diversifying the representation of structural information. Future research could focus on more deeply integrating these strategies to enhance the expressiveness of multi-hop processing paradigms within the context of dynamic hierarchy construction.

Table 11: Evaluation of DHMP with three independent tests.

| Model | Cylinder | Airfoil | Flag | Plate |
|-------|----------|---------|------|-------|
| RMSE-1 ($\times10^{-2}$) | 0.1506 $\pm$3.6E-4 | 36.27 $\pm$5.7E-4 | 0.2741 $\pm$2.4E-2 | 0.0263 $\pm$5.6E-6 |
| RMSE-All ($\times10^{-2}$) | 6.317 $\pm$0.33 | 2018 $\pm$130 | 68.66 $\pm$2.9 | 1.327 $\pm$0.002 |

Table 12: Full quantitative results over three training seeds.

| Model | Cylinder | Airfoil | Flag | Plate |
|-------|----------|---------|------|-------|
| RMSE-1 ($\times10^{-2}$) | | | | |
| MGN | 0.4046 $\pm$1.08E-2 | 77.38 $\pm$1.34E+1 | 0.4890 $\pm$6.34E-2 | 0.0579 $\pm$2.64E-3 |
| BSMS-GNN | 0.2263 $\pm$4.39E-2 | 71.69 $\pm$1.41E+1 | 0.5080 $\pm$0.48E-2 | 0.0632 $\pm$14.3E-3 |
| Lino *et al.* | 3.9352 $\pm$11.3E-2 | 85.66 $\pm$0.35E+1 | 0.9993 $\pm$2.44E-2 | 0.0291 $\pm$0.19E-3 |
| HCMT | 0.9190 $\pm$61.2E-2 | 48.62 $\pm$0.51E+1 | 0.4013 $\pm$1.76E-2 | 0.0295 $\pm$3.45E-3 |
| DHMP | **0.1568** $\pm$**0.94E-2** | **41.41** $\pm$**0.66E+1** | **0.3049** $\pm$**6.34E-2** | **0.0282** $\pm$**2.65E-3** |
| RMSE-All ($\times10^{-2}$) | | | | |
| MGN | 59.78 $\pm$2.00E+1 | 2816 $\pm$1.99E+2 | 124.5 $\pm$1.30E+1 | 3.982 $\pm$1.14E-2 |
| BSMS-GNN | 16.98 $\pm$0.12E+1 | 2493 $\pm$1.70E+2 | 168.1 $\pm$0.65E+1 | 1.811 $\pm$0.42E-2 |
| Lino *et al.* | 27.60 $\pm$0.86E+1 | 2080 $\pm$0.39E+2 | 118.2 $\pm$0.58E+1 | 2.090 $\pm$13.2E-2 |
| HCMT | 23.59$\pm$1.38E+1 | 3238 $\pm$3.62E+2 | 90.32 $\pm$0.50E+1 | 2.468$\pm$42.4E-2 |
| DHMP | **6.571** $\pm$**0.06E+1** | **2002** $\pm$**1.02E+2** | **76.16** $\pm$**1.30E+1** | **1.296** $\pm$**1.14E-2** |

## E    STABILITY ANALYSIS

Given the inherent randomness introduced by the Gumbel-Softmax sampling process in `DiffSELECT`, we evaluated the stability of DHMP by running the trained model on the test set in three independent trials. We report the mean and standard deviation of the prediction errors in Table 11. Despite the stochastic nature of the node selection process, the results show a very small standard deviation, demonstrating that DHMP reliably constructs stable and consistent dynamic hierarchies. This stability can be attributed to the `DiffSELECT` operation, where the node update module $\phi^v$ generates probabilities for retaining nodes in the next-level graph based on anisotropic aggregated edge features. The Gumbel-Softmax technique, coupled with temperature annealing, enables differentiable and stable node selection across hierarchy levels. As a result, the dynamic hierarchies are constructed in a manner that is not only consistent but also optimized for long-range dependencies. Moreover, the prediction errors from DHMP are significantly smaller than those of the baseline models, underscoring the robustness and reliability of the model, even with its dynamic node selection mechanism.

## F    FULL RESULTS OVER MULTIPLE TRAINING SEEDS

In Table 2 in the main manuscript, we report the mean results calculated over three random seeds. Here, in Table 12, we provide full comparisons between our model and the baseline models, including standard deviations.

## G    COMPUTATION EFFICIENCY

We evaluate computational efficiency based on four criteria: training cost required to reach model convergence, number of epochs/steps for model convergence, inference time per step, and the total number of model parameters. A performance improvement threshold of less than $1\%$ is used as the criterion for model convergence. The results are presented in Table 13.

Table 13: The detailed measurements of computation efficiency for DHMP and baseline models.

| Measurements | Dataset | MGN | BSMS-GNN | HCMT | DHMP |
|---|---|---|---|---|---|
| Training cost (hrs) | Cylinder | 35.26 | 37.11 | 80.60 | 35.96 |
| | Airfoil | 92.82 | 79.09 | 114.32 | 75.45 |
| | Flag | 28.12 | 18.27 | 66.70 | 17.14 |
| | Plate | 61.82 | 39.80 | 99.84 | 41.85 |
| Converged epochs \| steps | Cylinder | 31 \| 0.58M | 28 \| 0.52M | 32 \| 0.60M | 28 \| 0.52M |
| | Airfoil | 45 \| 0.84M | 39 \| 0.73M | 41 \| 0.77M | 39 \| 0.73M |
| | Flag | 35 \| 0.44M | 31 \| 0.39M | 37 \| 0.46M | 30 \| 0.37M |
| | Plate | 37 \| 0.46M | 26 \| 0.32M | 33 \| 0.41M | 28 \| 0.35M |
| Infer time/step (ms) | Cylinder | 17.35 | 16.55 | 79.52 | 21.79 |
| | Airfoil | 50.67 | 38.04 | 106.34 | 58.84 |
| | Flag | 16.15 | 17.18 | 85.87 | 26.33 |
| | Plate | 38.98 | 28.44 | 100.78 | 47.45 |
| #Parameter | Cylinder | 2.79M | 2.05M | 2.03M | 2.66M |
| | Airfoil | 2.79M | 2.58M | 2.03M | 2.27M |
| | Flag | 2.80M | 2.06M | 2.03M | 2.67M |
| | Plate | 2.80M | 2.87M | 2.03M | 3.20M |

## H  CONSTRUCTED DYNAMIC HIERARCHIES

We visualize the constructed context-aware and temporally evolving hierarchies in Figure 10. We can see that the constructed hierarchies evolve as the input context changes and the evolving graph structures align with high-intensity regions. We also visualize how the graph structure evolves across the entire sequence, shown in the GIF files in the supplementary.

## I  ROLLOUT ERRORS

Figures 11–13 showcase rollout error maps for the Airfoil, Flag, and DeformingPlate datasets. DHMP exhibits much lower rollout errors compared to the baseline models.

## J  DISCUSSION ON RELATED WORK

### J.1  COMPARISON TO STATIC HIERARCHICAL GNNs

Hierarchical GNNs with multi-level structures (Lino et al., 2022; Cao et al., 2023; Yu et al., 2024; Garnier et al., 2024; Hy & Kondor, 2023) are closely related to our approach, as they incorporate MPNNs within the U-Net architecture (Ronneberger et al., 2015). However, these methods typically treat multi-level structures as fixed preprocessing steps and do not adapt the graph hierarchies under varying physical conditions.

Besides, for intra-level feature propagation, some approaches use uniform feature aggregation (Cao et al., 2023; Lino et al., 2022; Hy & Kondor, 2023), while others employ attention mechanisms to introduce isotropic contributions from neighboring features (Yu et al., 2024; Garnier et al., 2024). However, the latter primarily focuses on adding weighted attention scores to nodes, overlooking spatially-aware edge features. For inter-level feature propagation, these methods typically rely on graph convolution based on node degree or directly adopt the U-Net architecture, limiting the flexibility in transferring information across hierarchical levels.

Different from our approach, Lino et al. (2022) uses manually set grid resolutions and spatial proximity for graph pooling, which requires manual hyper-parameters. BSMS-GNN (Cao et al., 2023) introduces a bi-stride pooling strategy that pools nodes on alternating breadth-first search frontiers while enhancing edges with two-hop connections. HCMT (Yu et al., 2024) refines the structure further by applying Delaunay triangulation to bi-stride nodes.

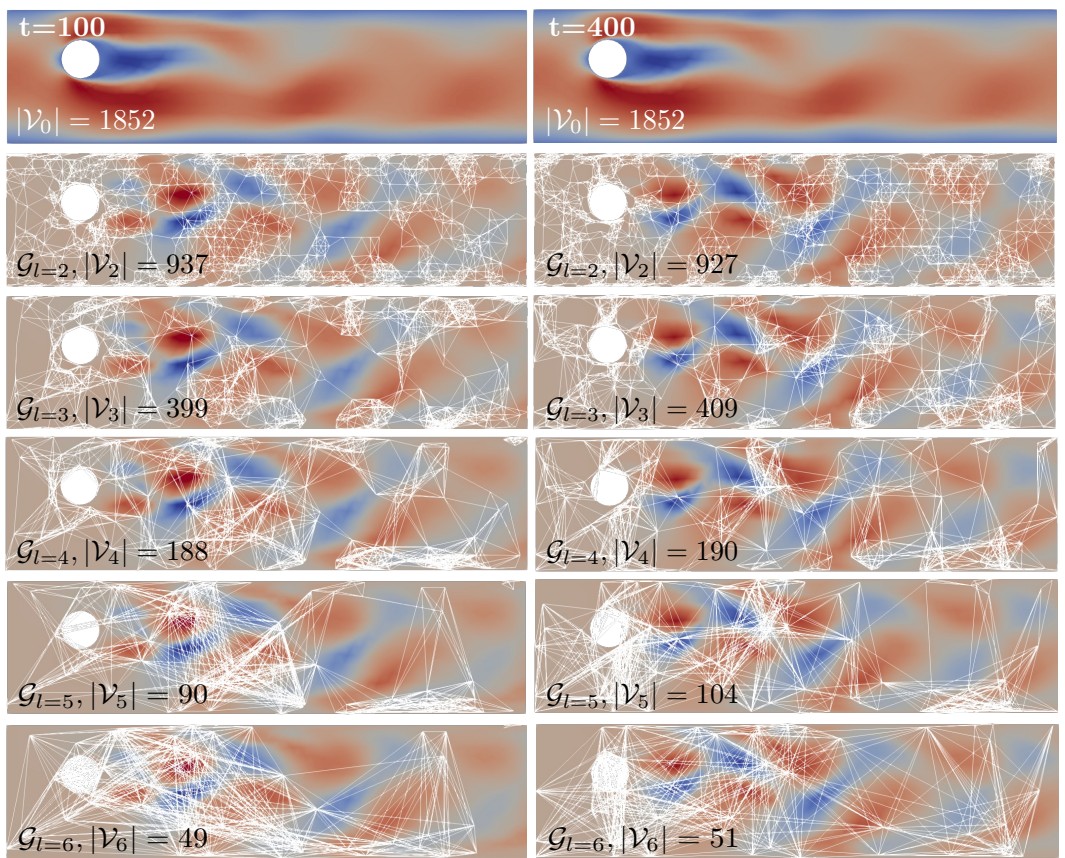

Figure 10: **Row 1:** The velocity field from the true data on the CylinderFlow dataset. **Row 2-6:** The temporal difference of the velocity fields between adjacent time steps alongside the constructed coarser-level graphs.

## J.2 COMPARISON TO DYNAMIC HIERARCHICAL GNNs

Recent literature has proposed methods to pool graphs into coarser-level representations (Hy & Kondor, 2023; Garnier et al., 2024). MGVAE (Hy & Kondor, 2023) employs the Gumbel-Softmax operation to partition the graph into discrete clusters at each resolution level, using a fixed $K$ value specifically for molecular graph generation tasks. However, this approach can be challenging for large graphs, as selecting an appropriate $K$ value may not scale well. Multigrid-GNN (Garnier et al., 2024), a concurrent work to our DHMP, introduces self-attention blocks to retain the top $k$ nodes at the coarse level and utilizes attention mechanisms to model intra-level feature propagation. However, both of these methods overlook inter-level feature transitions, primarily relying on the U-Net architecture without addressing the flexible exchange of information across different levels. In contrast, DHMP utilizes anisotropic message passing, which aggregates neighboring features in a

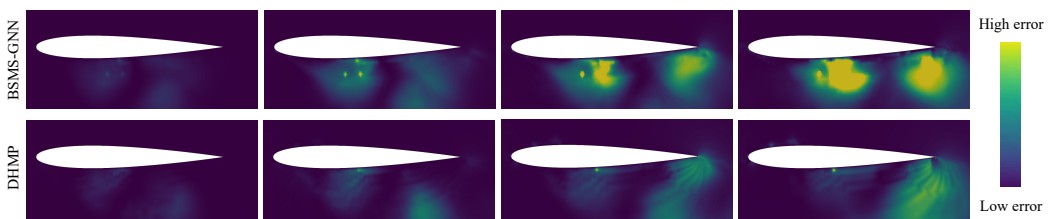

Figure 11: Showcases of rollout prediction error maps on Airfoil dataset.

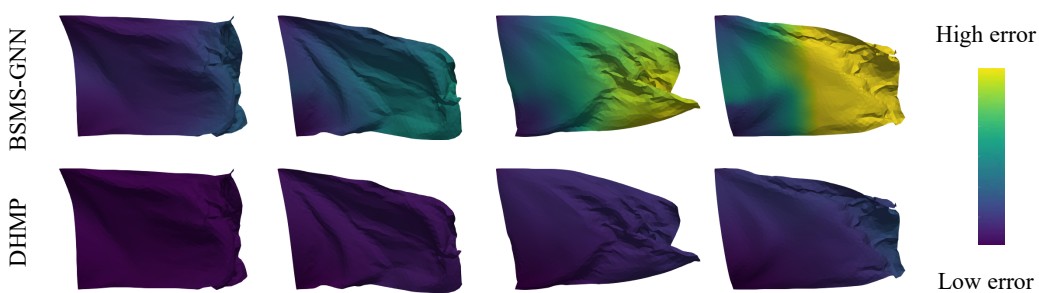

Figure 12: Showcases of rollout prediction error maps on Flag dataset.

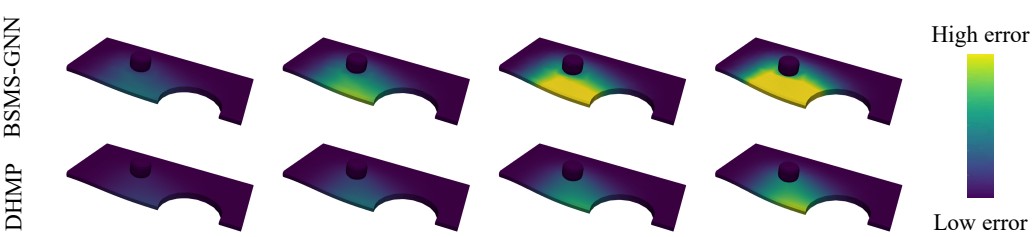

Figure 13: Showcases of rollout prediction error maps on DeformingPlate dataset.

directionally non-uniform manner. This approach allows DHMP to efficiently transfer information between levels by reusing importance weights, thereby overcoming the limitations of previous methods. Moreover, DHMP's differentiable node selection, predicted by the AMP block, eliminates the need for hyperparameter tuning and enables more flexible hierarchy construction. This dynamic approach offers significant advantages, especially in handling complex, large-scale graphs and facilitating inter-level feature propagation in dynamic simulations.

