# OpenReview forum: "Discovering Message Passing Hierarchies for Mesh-Based Physics Simulation"
_ICLR.cc/2025/Conference — Submitted to ICLR 2025_

### Official Review · Reviewer_pUWP · 2024-10-25

**Soundness:** 3
**Presentation:** 3
**Contribution:** 2
**Rating:** 5
**Confidence:** 5

**Summary:**

This paper introduces a novel multigrid method to coarsen and refine meshes dynamically, depending on the physics at the current time step. They also improve the usual sum aggregation in Message Passing by computing importance weights between each pair of nodes. They demonstrate their method on 4 usual physics datasets and a new one with adaptative re-meshing.

**Strengths:**

- a novel multigrid method based on Gumbel-Softmax sampling with only very few modifications to the graph net block (an extra probability output). The use of $K$-hop edges is also interesting.
- a new AMP layer that replaces the usual sum as the aggregation method in the node update with a weighted sum.
- good results improvements on meaningful and standard benchmarks
- a clever usage of the previously defined important to improve on the fine mesh interpolation method

**Weaknesses:**

- Important training details such as the number of training steps and the precise learning rate schedule are missing. At the moment, results from MeshGraphNet reproduced in the paper are much higher than in the original paper [1]. In my opinion, this hurts the paper a lot for two reasons: Was MGN (and the other models as well) simply undertrained to make the comparison with the new method better? If every model was actually properly trained for the right amount of steps, what would the comparisons look like now?
- missing a comparison with [2] that seems to follow a similar strategy with attention based on both the nodes updates, and the node selection during the coarsening stage
- l202-3-4: I think that statement is wrong. GAT networks for example do use edge features during the attention computation. Similarly, even with absolute node positions in the node features, why would an attention based method not be able to compute the differences between those coordinates?

**Questions:**

- l69-78: Can a parallel be made with [3] where each hop gets processed by a different MLP?
- l74/75: Where is this computational efficiency demonstrated?
- About the AMP layers: I am not sure I understand what improvements you gain in comparison to computing attention.
For the sake of clarity, I'll assume all functions are learnable parameters and let
  - $(v_i,v_j,e_{ij}) = \varepsilon_{ij}$
  - $\hat e_{ij} = W_1 \varepsilon_{ij}$
  - $\alpha_{ij} = \sigma (W_2 \varepsilon_{ij})$

With your current formulation, you have: $\sum \alpha_{ij} \hat e_{ij} = \sum \sigma (W_2 \varepsilon_{ij}) W_1 \varepsilon_{ij}$. This is very similar to an unscaled dot-product, but presented as a novelty. Can you specify why you chose to use such a method?
- What's the increase in computational complexity when using AMP instead of a regular GraphNetBlock?
- l213: How long does it take to construct such edges? What's the impact on the memory?
- Appendix Section D2 : You compare performances but not training/inference time and vRAM usage for the different $K$-values. It would be very interesting to add those.

**Changes**

- l108: You should specify that this feature propagation is related to the edge's length, not their number per se.
- l115: "However..." : issue with the sentence
- l152: "where..." : issue with the sentence
- Section 4: You define the number of layers for MGN but not the architecture details for the other models.
- Table 2: It might be worth it to put in \emph or with another strategy the second best result.
- l761: the inflow velocity varies as well
- table 7 l791: There's a typo for the noise used in the Airfoil Dataset
- l807: there's a typo for $n$ the node type

**Additional comments**

- Figure 1, I am unsure about the usefulness of the gradient arrows.
- Section 5 could benefit from presenting [4]

[1] : Learning Mesh-Based Simulation with Graph Networks

[2] : Multi-Grid Graph Neural Networks with Self-Attention for Computational Mechanics

[3] : How Powerful are K-hop Message Passing Graph Neural Networks

[4] : GraphCast: Learning skillful medium-range global weather forecasting

---

> ### Author Response · Authors · 2024-11-22
> **Responses to Reviewer pUWP (Part 1)**
>
> We appreciate the reviewer’s detailed comments and hope that the following responses can adequately address the reviewer’s concerns. We are more than willing to address any further questions you may have.
>
> > **W1.** Experiment details and performance discrepancy.
>
> Please refer to our **General Response**.
>
> > **W2.** Missing comparison with [2].
>
> We thank the reviewer for bringing this paper to our attention. First, we would like to kinldly remind the reviewer that, according to the ICLR reviewer guidelines: "authors are not expected to compare with work published within four months of the paper deadline". Since [2] was uploaded to arXiv just two weeks before the deadline and has not yet been officially published, it should be considered concurrent with our work.
>
> Nevertheless, we recognize the value of such a comparison and outline the key differences from [2] below. We have also included the discussion in Appendix J.
> - First, DHMP assigns different importance weights to the updated edge features, highlighting the anisotropic influence of these edges, while Garnier et al. primarily focus on attention-based interactions between updated node features.
> - Second, DHMP constructs dynamic hierarchies based on the differentiable node selection achieved through AMP's predictions, while Garnier et al. introduce additional self-attention blocks to retain the top $k$ nodes.
>
> > **W3.** l202-3-4: I think that statement is wrong. GAT networks for example do use edge features during the attention computation. Similarly, even with absolute node positions in the node features, why would an attention-based method not be able to compute the differences between those coordinates?
>
> Sorry for the confusion. We have revised the manuscript to clarify this point.
>
> Some attention-based GNNs, such as HCMT (Han et al.), do incorporate edge features when computing attention weights. However, in these methods, the resulting weights are typically applied to aggregate node features. In contrast, our proposed AMP directly applies the predicted importance weights to the edge features themselves and, more importantly, further exploits these weights in inter-level feature propagation.
>
> Furthermore, while attention-based methods for physical simulations (e.g., HCMT and GMR-Transformer-GMUS) rely on transformer architectures with quadratic time complexity, AMP maintains a computational complexity comparable to that of a standard GraphNetBlock. This makes AMP more computationally efficient and effective for physical simulations.
>
> > **Q1.** l69-78: Can a parallel be made with [3] where each hop gets processed by a different MLP?
>
> Yes, indeed! We have included this result in Appendix D.5 in the revised paper. We separate the $\mathcal{E}_{l}$ and $K$-hop enhanced edges into distinct variables to align with the approach in [3]. We replace the AMP block in DHMP with the KP-AMP block. We conduct experiments on CylinderFlow and present the results below. We can see that incorporating distinct MLPs for each hop helps to capture varying structural information at different scales. This could be a promising direction for further improvement upon DHMP.
>
> ||RMSE-1(× 10⁻²)| RMSE-All(× 10⁻²)|
> |-|-|-|
> |DHMP | 0.1568|6.571|
> |DHMP w/ KP-AMP |0.1438|6.444|
>
> > **Q2.** l74/75: Where is this computational efficiency demonstrated?
>
> Table 13 in Appendix G presents the training cost and inference time for baseline models and DHMP. The results indicate that HCMT (Yu et al.), which incorporates an attention mechanism in its message passing, requires significantly more training hours to reach convergence and exhibits a substantially higher inference time, demonstrating its greater computational overhead. Specifically, HCMT requires 1.5x to 3.9x training cost and 1.8x to 3.7× inference time compared to DHMP.

---

> ### Author Response · Authors · 2024-11-22
> **Responses to Reviewer pUWP (Part 2)**
>
> > **Q3.** About the AMP layers.
>
> We use AMP over traditional attention mechanisms for its better performance in dynamic systems. First, we have compared the results of DHMP with HCMT, which employs an attention-based GNN architecture. As shown in Table 2, DHMP outperforms HCMT in all cases.
>
> Second, we indeed considered using scaled-dot product attention instead of the AMP module by replacing the original edge update function with a cross-attention that integrates both edge features and neighboring node features. Specifically, the edge feature is updated as $\mathbf{\hat{e}}\_{aggr} = \operatorname{CrossAttention}(Q = \mathbf{v}\_i, \, K = \{\hat{\mathbf{e}}\_{ij}\}, ,V = \{\hat{\mathbf{e}}\_{ij}\})$, where the edge feature is refined by attending to node features. The cross-attention scores are then reused for inter-level information propagation. The results are as follows:
>
> |                     | Cylinder |          | Flag   |          |
> | ------------------- | -------- | -------- | ------ | -------- |
> | RMSE (× 10⁻²)     | RMSE-1   | RMSE-All | RMSE-1 | RMSE-All |
> | DHMP-CrossAttention | 0.1881   | 8.35     | 0.3650 | 85.12    |
> | DHMP-AMP            | 0.1568   | 6.571    | 0.3049 | 76.16    |
>
> The results demonstrate that DHMP with AMP outperforms the cross-attention variant in overall accuracy. This is likely because the edge features $\hat{\mathbf{e}}\_{ij}$ inherently incorporate information from the associated nodes $\mathbf{v}\_i$, making it unnecessary to explicitly include $\mathbf{v}\_i$ in the computation, which would introduce redundancy.
>
> > **Q4.** Increased computational complexity by using AMP instead of GraphNetBlock.
>
> The increased computational complexity when using AMP instead of a regular GraphNetBlock comes primarily from the addition operation of importance weights and the softmax normalization.
>
> For both GraphNetBlock and AMP, given the $\mathbf{e}\_{ij}$ with constant dimension $c$ and $\hat{\mathbf{e}}\_{ij}$ and $\mathbf{v}\_{i}$ with dimension $D$, the edge update and node update functions have computational complexities of $O(|E| \cdot (cD + D^2))$ and $O(|E| \cdot D + |V| \cdot D^2)$. AMP computes the importance weight by $\phi^w$, which incurs an additional complexity of $O(|E| \cdot (cD + D^2))$. Furthermore, the softmax operation results in an additional complexity of $O(|E| \cdot D)$.
>
> Although the computation of the importance weights doubles the computational cost per edge compared to the regular GraphNetBlock, the overall complexity still remains $O(|E|D + |V|D^2)$.
>
> In the following table, we replace the GraphNetBlock in MGN with AMP to illustrate the increased computational cost (data: CylinderFlow).
>
> |   | Training time (ms) | Traning RAM (GBs)| Infer time (ms)| Infer RAM (GBs)|
> |-|-|-|-|-|
> | MGN| 26.05 | 21.97 | 17.35 | 1.17|
> | MGN+AMP | 32.63 | 28.60| 23.37|1.18|
> | DHMP | 28.78 | 17.32| 21.79| 1.19 |
>
> > **Q5&6.** Time & memory cost of the construction of K-hop edges. Appendix D2: The training/inference time and vRAM usage for the different K-values.
>
> In the revised Table 9 in Appendix D.2, we present comparisons of training/inference time and vRAM usage for edge construction of different $K$-values on the FlagSimple dataset. As $K$ increases, both the training and inference times gradually rise due to the enhanced edge construction, which involves more computational overheads.
>
> > **Changes and additional comments.**
>
> Thank you for your careful checking and constructive feedback. We have revised our paper accordingly in L108, L116, L152, Table 2, L761, and L807. Additionally, we have included reference [4] in the related work and presented more details of the model architectures in Appendix C.

---

> > ### Comment · Reviewer_pUWP · 2024-11-23
> >
> > Thank you for your answers!
> >
> > Regarding W1: similar to reviewer afN4, I still have a lot of doubts regarding the training pipeline you are using. Even with the edge encoding:
> > - the results shown are still much higher than what was previously seen
> > - the difference between your methods and MGN is much smaller
> >
> > Those 2 points make me think that with the right training procedure, that gap could be even smaller (or even reversed?). The paper would greatly benefit from being able to reproduce the MGN metrics from the original paper and then use the same pipeline for your approach.
> >
> > Regarding W2: thanks for the clarification. Also, very sorry I completely forgot the ICLR guidelines regarding paper recency.
> >
> > Regarding Q3: is the overall RMSE on the Flag dataset a typo? Otherwise it seems that in this case the cross attention is better (or at least not statistically different from your approach)
> >
> > Again, thanks for all the clarifications. At the moment (mainly due to W1), I am not ready to update my rating of the paper.

---

> > > ### Author Response · Authors · 2024-11-24
> > >
> > > Thank you for your time and quick response!
> > >
> > > Regarding W1: Please refer to the new comment in the general response.
> > >
> > > Regarding Q3: This is a typo. We've corrected the result in the reply.

---

### Official Review · Reviewer_rDem · 2024-10-28

**Soundness:** 3
**Presentation:** 3
**Contribution:** 3
**Rating:** 6
**Confidence:** 3

**Summary:**

Summary: The paper presents a new approach to learning to simulate physical systems using Graph Neural Networks (GNNs). Traditional GNNs rely on fixed, manually designed hierarchies / meshes, which fail to adapt to the evolving dynamics in physical simulations. The authors propose the Dynamic Hierarchical Message Passing (DHMP) model, which introduces dynamic, context-aware and data-driven hierarchies.

Key innovations of DHMP include:
* Anisotropic Message Passing: Facilitates direction-specific message propagation, allowing better representation of physical processes.
* Differentiable Node Selection: This component allows for learning adaptable, multi-scale graph structures that evolve over time.

DHMP outperforms existing methods, achieving an average of 22.7% improvement in five classic physics simulation datasets. It effectively models both local and long-range dependencies in time-varying, mesh-based systems.

**Strengths:**

The Dynamic Hierarchical Message Passing (DHMP) model adapts its graph structure dynamically, effectively captures long-range dependencies and handles unseen mesh structures, making it a strong solution for complex physics simulations.

**Weaknesses:**

Some potential limitations:

* Novelty of multi-scale graph neural networks by differentiable node selection: This work "Multiresolution equivariant graph variational autoencoder" by Truong Son Hy and Risi Kondor (https://iopscience.iop.org/article/10.1088/2632-2153/acc0d8) has already proposed a similar idea using Gumbel-Softmax for node sampling to construct an adaptive hierarchy.

* Increased Complexity: Theoretically, the dynamic adaptation of hierarchies and anisotropic message passing introduces additional computational overhead, making it more complex and potentially slower than fixed-hierarchy models. Could you please analyse the time complexity and the space complexity of your model and compare with other baselines? In the Appendix, Table 13 includes comparison with other baselines in terms of training cost, inference time and number of parameters that suggest the computational overhead of this work is not significant.

* Stability of Differentiable Node Selection (DiffSELECT): This learning mechanism might face instability challenges during training, especially in highly dynamic or chaotic systems, which could lead to less reliable performance in certain scenarios. The key component / function is the Gumbel-Softmax in the node sampling / selection (see Equation 6 in Section 3.3). However, the Gumbel-Softmax is sensitive with its temperature hyper-parameter \tau (see PyTorch instruction: https://pytorch.org/docs/stable/generated/torch.nn.functional.gumbel_softmax.html). How can you select the temperature hyper-parameter? Is it the same for every scenario?

It would be great if the authors can try on some turbulence datasets to showcase the stability of method.

* Limitation in Generalization: While DHMP performs well on some specific physics simulation datasets, its generalization to other domains or non-mesh-based applications may require further modification or tuning, limiting its broader applicability. Do you have any plan to apply your model into other domains or non-mesh-based applications?

I suggest the authors to check PDEBench benchmark: https://arxiv.org/abs/2210.07182

**Questions:**

I would like the authors to address the potential limitations that I have listed.

---

> ### Author Response · Authors · 2024-11-22
> **Responses to Reviewer rDem (Part 1)**
>
> Thank you for your valuable comments, and we hope that the added results and the clarification below can address your concerns. For any further questions, please don’t hesitate to contact us.
>
> > **W1.** Novelty of multi-scale graph neural networks with differentiable node selection.
>
> Thank you for bringing our attention to this related work. We have included proper citations and detailed comparisons in the revised version (Sec 5 and Appendix J). Below, we outline the key differences between MGVAE and our work.
> - **Usage of Gumbel-Softmax operation.** The Gumbel-Softmax operation is used differently in MGVAE, where it partitions the graph into discrete clusters with a specified $K$ value at each resolution level. This approach may not be suitable for large graphs, as choosing appropriate and high $K$ values can be challenging. In contrast, DHMP uses Gumbel-Softmax for differentiable node selection without requiring hyperparameter selection.
> - **Edge construction at coarse levels.** While MGVAE constructs edges at coarser levels by aggregating adjacency information within clusters, DHMP constructs edges using a $K$-enhanced edge set that captures long-range dependencies.
> - **Intra/Inter-level feature propagation.** DHMP employs anisotropic message passing to aggregate neighboring features in a directionally non-uniform manner, reusing importance weights to enable efficient inter-level information propagation. However, MGVAE primarily relies on GraphNetBlocks for message passing, which builds a bottom-up aggregation and top-down decoding strategy, pooling features at each level and projecting them across the hierarchy in a variational manner.
>
> Below, we highlight the key differences between the two methods:
>
> |Aspect |DHMP |MGVAE  |
> |-|-|-|
> | Primary Application  | Physics-based dynamic simulations | Molecular and general graph generation                                                                        |
> |Graph Structure Construction | Dynamic hierarchy with differentiable, context-aware node selection   | Predict graph clustering, specifying $K$ clusters per level |
> | Role of Gumbel-Softmax | Used for dynamic node selection, enabling adaptive hierarchy  | Tailored for $K$-clustering to partition nodes at each hierarchical level  |
> | Edge Construction at Coarse Levels | Construct $K$-enhanced edge set | Aggregate adjacency information within clusters |
> | Inter-Level Feature Propagation| Use AMP weights to propagate inter-level feature | Bottom-up aggregation and top-down decoding, pooling features in clusters for encoding and projecting in decoding in a variational manner|
> | Performance Focus | Long-term physical predictions, modeling space-time dynamics efficiently| High fidelity in generating hierarchical latent representations for static graph generation tasks |
>
> > **W2.** Complexity analysis.
>
> The asymptotic complexity of our approach is $O(|E|d + |V|d^2)$, which is consistent with standard GNNs. For baselines, the message passing GraphNetBlock with MLP implementation introduces a time complexity $O(|E|d + |V|d^2)$, where $|E|$ is the number of edges and $|V|$ is the number of nodes, $d$ is the feature dimension. Methods like MGN process message passing on high-resolution meshes, resulting in higher computational overhead. In contrast, hierarchical methods like BSMS-GNN reduce computation since coarser graph levels have fewer edges and nodes. In our proposed DHMP, the additional operations primarily come from the **dynamic graph construction** and **anisotropic message passing**, with detailed complexity as follows.
> - **Dynamic Graph Construction:** (1) Probability-based node selection involves computing probabilities for all nodes, resulting in a complexity of $O(|V|)$. (2) After node selection, edges are pooled to form a coarsened graph. This operation involves evaluating and updating edges for the selected nodes, with a complexity of $O(|E|)$.
> - **Anisotropic Message Passing:** AMP introduces an overhead for predicting directional weights on edges. We compute the directional weight and perform weighted aggregation, maintaining a complexity of $O(|E|d + |V|d^2)$.
>
> Overall, the asymptotic complexity of DHMP is $O(|E|d + |V|d^2)$, which is consistent with standard GNNs. The computation of the importance weights doubles the computational cost per edge compared to the regular GraphNetBlock. Therefore, the computational overhead is acceptable.

---

> ### Author Response · Authors · 2024-11-22
> **Responses to Reviewer rDem (Part 2)**
>
> > **W3.** Stability of Differentiable Node Selection (DiffSELECT) and sensitivity to temperature hyper-parameter \tau.
>
> **(1) The influence of Gumbel-softmax sampling on performance stability.**
>
> We showcase the stability of our model in three aspects:
> - **Test stability:** As shown in Table 11 in Appendix E, DHMP provides stable results under independent tests with Gumbel-Softmax sampling.
> - **Training stability:** In Table 12 in the revised Appendix F, we present a full comparison between our model and the baseline models, calculated across three random seeds. The shown standard deviations highlight the advantage of DHMP in maintaining stable training.
> - **Robustness to out-of-distribution systems:** To further evaluate stability in highly dynamic systems, we have evaluated DHMP on datasets with significantly higher velocity norms and variances (Table 5 in Sec 4.4). As shown below, the standard deviation of prediction errors remains minimal, highlighting that once trained, DHMP consistently produces stable predictions under Gumbel-softmax sampling, even in highly dynamic scenarios.
>
> |      | Cylinder  (× 10⁻²)        |                   | Airfoil   (× 10⁻²)         |                  |
> | ---- | --------------- | ----------------- | ---------------- | ---------------- |
> |      | RMSE-1| RMSE-ALL | RMSE-1   | RMSE-ALL |
> | DHMP | 0.214±5.99E-6   | 9.10±1.09E-2      | 66.6±9.31E-4     | 2260 ± 2.39      |
>
> **(2) Sensitivity to the Gumbel-Softmax temperature during training.**
>
> We acknowledge the sensitivity of the Gumbel-Softmax to temperature ($\tau$). In our experimental setting, we employed temperature annealing during training, gradually decreasing the temperature from 5 to 0.1 with exponential decay on all datasets, as detailed in Appendix B (lines 817-819). This approach balances exploration in early training with progressively refined node selection afterwards, enhancing both stability and performance while reducing the need for hyperparameter tuning [1-2]. Table 12 in Appendix F shows the results of the mean and standard deviation calculated over three random training seeds under this setting.
>
> Additionally, we have experimented with fixed temperatures ($\tau = 0.1, 0.3, 0.5$) on the CylinderFlow dataset. The results are shown below:
>
> | $\tau$  | RMSE-1(× 10⁻²) | RMSE-All (× 10⁻²)|
> | -------------------- | ------ | -------- |
> | 0.1                  | 0.312  | 7.220    |
> | 0.3                  | 0.207  | 9.524    |
> | 0.5                  | 0.297  | 9.287    |
> | Annealing: 5 to 0.1 | 0.157  | 6.571    |
>
> [1] Categorical reparameterization with gumbel-softmax.
>
> [2] The concrete distribution: A continuous relaxation of discrete random variables.

---

> ### Comment · Area_Chair_w26o · 2024-11-27
> **Rebuttal Response**
>
> Dear Reviewer,
> Do you mind letting the authors know if their rebuttal has addressed your concerns and questions? Thanks!
> -AC

---

> ### Author Response · Authors · 2024-11-28
>
> Dear rDem,
>
> Thank you once again for your review of our work. As the discussion period is approaching its end, we would be grateful if you could reply whether our responses have addressed your questions and concerns. Your time and effort are greatly valued.

---

### Official Review · Reviewer_afN4 · 2024-10-29

**Soundness:** 3
**Presentation:** 3
**Contribution:** 3
**Rating:** 5
**Confidence:** 4

**Summary:**

The paper proposes two improvements to hierarchical GNN-based neural physics simulator surrogates. The first improvement is an anisotropic message passing mechanism which replaces sum aggregation with a weighted softmax aggregation. The second improvement is a differentiable node selection mechanism to learn long-range dependencies.

**Strengths:**

- The method is well motivated, presented in a clear way with nice visualizations and a comprehensive analysis of each proposed improvement.
- An extensive evaluation is provided with additional out-of-distribution evaluations, which are interesting to see.

**Weaknesses:**

- While Figure 5 provides nice insights into high error regions, Figure 3 suggests that the model also assigns lots of nodes towards relatively uninformative regions. In the bottom right of the bottom left image of Figure, one can see a region that is densely populated with graph nodes, even though the flow is almost laminar in that region. A similar behavior can be observed in the bottom left region of the bottom right image of Figure 3. Figure 5 shows that the model tends to assign more nodes to challenging regions, but Figure 3 also suggests that lots of nodes are assigned to easy/uninformative regions.

- The number of hierarchies seems like an important hyperparameter, considering that the paper proposes to learn hierarchies instead of using static ones. A discussion how this hyperparameter is selected (e.g. based on problem size) and implications of unfavorable choices of this hyperparameter (e.g. via an ablation study) would strengthen the paper.

- MGN reports vastly different performance of their method for the considered benchmarks, often outperforming DHMP. For example, DHMP reports 0.414 RMSE-1 for DHMP on the Airfoil dataset while MGN reports 0.314 for MGN (where the submitted paper reports 0.7738 for MGN).

**Questions:**

- Is there an intuition for why it seems (as shown in Figure 3) that also uninformative regions get lots of graph nodes assigned?
- How would the distribution of nodes look over the whole dataset or a single simulation trajectory? E.g. are most nodes assigned to the region where turbulent flow happens instead of laminar flow?
- How is the number of hierarchy selected? Based on a validation set or based on the problem size/difficulty?
- What is the difference between the considered benchmarking settings and the benchmarks conducted in MGN? Why are the MGN paper results sometimes better than DHMP?

---

> ### Author Response · Authors · 2024-11-22
>
> We appreciate the reviewer’s detailed comments and hope that the following responses can adequately address the reviewer’s concerns. We are more than willing to address any further questions you may have during the discussion period.
>
>
> > **W1 & Q1:** Nodes distributed in uninformative regions; **Q2:** Node distribution over trajectories.
>
> **(1) "Uninformative" regions.**
>
> As a data-driven neural network, DHMP does not have access to physical priors, such as knowledge of where turbulence or laminar flow occurs. As a result, it may retain nodes in regions that could be perceived as "uninformative" from a physical standpoint.
>
> However, it's important to note that these uninformative regions, particularly those with laminar flow, are not necessarily trivial or easy. As illustrated in the error map at the bottom of Figure 3, baseline models like MGN and BSMS-GNN show higher errors in these regions, indicating that they present challenges for these models as well. In these challenging regions (even with laminar flow), excluding interactions from these areas entirely could hinder the model’s ability to learn a more comprehensive representation of the system dynamics. These regions can still influence other parts of the system. In other words, retaining these nodes in top hierarchies is crucial for a more complete understanding of the overall dynamics and long-range node relations.
>
> **(2) Node distribution across the entire sequence.**
>
> We have visualized the evolution of the graph structure across the entire sequence in the GIF files uploaded in the revised supplementary material. The results demonstrate that nodes are typically distributed in regions with larger temporal differences, supporting our observation that the model prioritizes areas with more  dramatic changes in fluid motion.
>
>
> > **W2&Q3.** About the number of hierarchies. How is the number of hierarchy selected? ... based on the problem size/difficulty?
>
> Thank you for the insightful question! The number of hierarchy levels in DHMP is generally determined based on the problem size, roughly $O(\log |V|)$. This choice is also consistent with static hierarchy methods such as BSMS-GNN.
>
> We acknowledge that the hierarchy parameter may impact model performance differently, so we conducted a comparison on the CylinderFlow dataset to analyze DHMP and BSMS-GNN under various hierarchy parameter settings. The results are as follows and have been included in **Figure 9** in Appendix D4.
>
> |          | RMSE-1 (× 10⁻²)|        |        |        | RMSE-All (× 10⁻²)|       |       |       |
> | -------- | ------ | ------ | ------ | ------ | -------- | ----- | ----- | ----- |
> | \#Hierarchy | 3      | 5      | 7      | 9      | 3        | 5     | 7     | 9     |
> | BSMS-GNN | 0.3634 | 0.2809 | 0.2263 | 0.2391 | 39.51    | 30.12 | 16.98 | 35.14 |
> | DHMP     | 0.2833 | 0.1892 | 0.1568 | 0.1618 | 16.44    | 7.497 | 6.571 | 8.456 |
>
> The results indicate that DHMP outperforms BSMS-GNN at all hierarchy levels. Both models benefit from increased hierarchy depth up to level 7, suggesting deeper levels enhance accuracy by capturing more complex interactions. However, a slight performance drop at level 9 hints at possible overfitting. Overall, the dynamic hierarchical models in DHMP consistently outperform their counterparts with static hierarchies in BSMS-GNN.
>
>
> > **W3 & Q4.** Discrepancy in model performance. Experiment setup and implemantation details.
>
> Please refer to our **General Response**.

---

> > ### Comment · Reviewer_afN4 · 2024-11-22
> >
> > Thank you for the clarifications!
> >
> > The GIF visualization is a bit hard to follow as it jumps in 50 timesteps increments. I think a more informative visuaization would be to show a heatmap of node frequency per position. Intuitively, the areas with transient flow should have higher frequency of nodes than "easy" areas. If also the laminar flow regions can be challenging, I think it would be instructive to compare the average error per node with the node frequency per node. This would be similarto Figure 5a but for a whole trajectory instead of a single frame in a trajectory.

---

> > > ### Author Response · Authors · 2024-11-24
> > >
> > > Thank you for your reply!
> > >
> > > The current visualizations (GIF files) effectively capture the relationship between error and the learned sampling probability within dynamic systems, as they illustrate how DHMP dynamically allocates nodes to regions of higher temporal variation. This captures transient dynamics and subtle shifts in fluid motion across individual timesteps.
> > >
> > > Per the reviewer's request, we visualized the **time-average** errors, temporal differences, and sampled frequencies(the frequency of nodes left in the last hierarchy) in the revised supplementary zip. Averaging node frequencies and error values over an entire sequence would dilute the fine-grained, timestep-specific variations that DHMP captures, which are critical to representing the adaptive nature of our model's node selection. These subtle shifts are essential to understanding how DHMP prioritizes transient regions and adapts to localized dynamics within the system.
> > >
> > > Thus, the visualization of the **evolution** of the graph structure across the entire sequence provides a more accurate and comprehensive demonstration of DHMP's response to dynamic changes, without oversimplifying or averaging out essential temporal variations.

---

### Official Review · Reviewer_JVHC · 2024-11-04

**Soundness:** 2
**Presentation:** 3
**Contribution:** 3
**Rating:** 5
**Confidence:** 4

**Summary:**

The paper proposes an anisotropic hierarchical message passing technique together with dynamic coarser graph construction for such a hierarchy implementation. The authors claim better simulation quality for a range of test cases.

**Strengths:**

* new implementation of attention-based and hierarchical message passing in GNNs
* learnable hierarchy
* generalization and ablation studies

**Weaknesses:**

* The proposed approach of anisotropic message passing differs only in small implementation details from graph attention and cannot be considered really novel
* In Table 2 the results from cited works are far from the ones reported in them. If one looks at the reported results in the MGN paper [33] Tobias Pfaff, Meire Fortunato, Alvaro Sanchez-Gonzalez, and Peter Battaglia. Learning mesh-based simulation with graph networks. In International Conference on Learning Representations, 2021, these results will be better than here.

also the paper is not structured well, the authors main contributions are not written separately

**Questions:**

Why there is a huge difference in Table 2 for MGN model results in your paper and in the original paper? (probably also for another cited works)?

---

> ### Author Response · Authors · 2024-11-22
>
> Thank you for your comments. We have made every effort to answer each of your questions, and hope that our response can address your concerns regarding this paper.
>
> > **W1.**  The proposed anisotropic message passing differs only in small implementation details from graph attention.
>
> We would like to clarify the novelty and contributions of our method, particularly addressing the role of AMP within the broader framework of dynamic hierarchy construction from the following aspects.
> - **Dynamic hierarchy construction.**  Firstly, we would like to emphasize primary contribution lies in constructing dynamic hierarchies that adapt to evolving physical dynamics. In this hierarchy-building process, AMP serves two key functions. First, it aggregates neighboring features to the central node in a directionally non-uniform manner. Second, it utilizes the importance weights predicted by AMP to enable efficient inter-level information propagation. The latter aspect has been unintentionally overlooked by the reviewer; however, our ablation study (Figure 4, lines 429–454) demonstrates its benefit on model performance.
> - **Differences between AMP and graph attention.** Unlike traditional graph attention, AMP incorporates nodes' relative positions directly into edge features, capturing essential spatial relationships that drive accurate modeling of physical systems where spatial orientation greatly affects physical dynamics. Graph attention computes attention scores between nodes, while AMP directly applies the predicted importance weights to edge features. By using anisotropic interactions through edge features that embed geometric information, our approach enhances generalizability and robustness.
> - **Experimental results.** We compared DHMP with HCMT, an attention-based baseline. As shown in Table 2 and Table 13, DHMP consistently outperforms HCMT in both performance and efficiency (especially training cost and inference time). Additionally, we explored replacing AMP with a cross-attention mechanism that integrates both edge and neighboring node features while reusing attention scores for inter-level feature propagation. This modification resulted in reduced performance, as detailed in our response to Reviewer 96Nr’s Q3.
>
> Overall, while AMP is a key component of our dynamic hierarchy framework, it is not the sole primary contribution of our method. We hope these clarifications further highlight the effectiveness and novelty of our approach.
>
>
> > **W2 & Q1.** Differences in results.
>
> Please refer to our **General Response**.
>
> > **The authors' main contributions are not written separately.**
>
> Thank you for your suggestion. We have revised the introduction section and listed the contributions of our work. Please feel free to share any further suggestions for improving the paper.

---

> ### Comment · Area_Chair_w26o · 2024-11-27
> **Rebuttal Response**
>
> Dear Reviewer,
> Do you mind letting the authors know if their rebuttal has addressed your concerns and questions? Thanks!
> -AC

---

> > ### Comment · Reviewer_JVHC · 2024-11-28
> > **Answer and rating increase**
> >
> > Thanks to the authors for their persistent work on the reviewers' comments and questions!
> >
> > Following the paper revision and the discussion with the other reviewers I am convinced that I can increase:
> > Presentation: 2->3
> > Contribution: 2->3
> > Rating 3->5
> >
> > As for me, the final comparison of the results still is not absolutely convincing. However the existing approach can be for sure considered interesting and could benefit some part of the ML physics simulation community. I didn't increase the grades even more because I also think that the proposed method has very limited practical applicability (it's generalizability is far from the best as seen in the paper, also it wasn't tested on complex cases with big graphs) and speaking about its novelty I think it's rather limited.
> >
> > However I think the paper has the potential to be published in a good venue after conducting some additional evaluation and finding the practical cases in which the method will work the best.

---

> > > ### Author Response · Authors · 2024-11-28
> > >
> > > Dear JVHC,
> > >
> > > thank you for increasing your score! We appreciate your careful review and constructive feedback.

---

### Meta-Review · Area_Chair_w26o · 2024-12-19

**Metareview:**

**Summary** This paper studies the program of simulating physical systems over meshes using hierarchical graph neural networks.  The proposed method, Dynamic Hierarchies for Message Passing (DHMP), has two innovations: anisotropic aggregations using a weighted softmax and dynamic graph hierarchies using differentiable node selection.

**Strengths** Reviewers appreciated the overall DHMP architecture and agreed the anisotropic aggregations and dynamic hierarchical approach were effective, cleverly used, and appropriate to the application, capturing long range dependencies and generalizing to new meshes. Experiments are performed over reasonable benchmarks. The paper has good ablations justifying the effectiveness of these components.

**Weaknesses** After discussion and rebuttal the main concern was on making a fair comparison to previous methods, namely mesh graph nets, which reports better performance in their own paper. Despite efforts by the authors to explain the differences in terms of the effect of edge encodings and wether pressure was also predicted and incorporated into the metric, several reviewers were not convinced.  An additional common concern among reviewers was the lack of novelty – the two main contributions, anisotropic message passing and soft node selection have appeared in related work in some form.

**Conclusion** The majority of reviewers felt that the weaknesses above were too significant to overlook.

**Additional Comments On Reviewer Discussion:**

After the revision, JVHC increased their score from 3 to 5 but did not go higher due to limited novelty and not finding the result comparison fully convincing.

---

### Decision · Program_Chairs · 2025-01-22

Reject